# Efficient high-precision homology-directed repair-dependent genome editing by HDRobust

Stephan Riesenberg [1] ✉, Philipp Kanis [1], Dominik Macak [1], Damian Wollny[1], Dorothee Düsterhöft[1], Johannes Kowalewski[1], Nelly Helmbrecht [1], Tomislav Maricic [1] & Svante Pääbo [1,2]

Homology-directed repair (HDR), a method for repair of DNA double-stranded breaks can be leveraged for the precise introduction of mutations supplied by synthetic DNA donors, but remains limited by low efficiency and off-target effects. In this study, we report HDRobust, a high-precision method that, via the combined transient inhibition of nonhomologous end joining and microhomology-mediated end joining, resulted in the induction of point mutations by HDR in up to 93% (median 60%, s.e.m. 3) of chromosomes in populations of cells. We found that, using this method, insertions, deletions and rearrangements at the target site, as well as unintended changes at other genomic sites, were largely abolished. We validated this approach for 58 different target sites and showed that it allows efficient correction of pathogenic mutations in cells derived from patients suffering from anemia, sickle cell disease and thrombophilia.

In CRISPR-mediated genome editing, CRISPR nucleases are used to introduce double-stranded breaks (DSBs) at genomic sites that are complementary to the spacer sequence of a guide (g)RNA[1,2]. Sometimes, DSBs can also occur at unintended 'off-target' sites that have sequence similarity to the target site[3]. Cellular repair of these DSBs often results in mutations and thus genome editing, while inability to repair the DSB will result in cell death[4]. DSB repair is mainly carried out by nonhomologous end joining (NHEJ) and by microhomology-mediated end joining (MMEJ), which can serve as a backup for NHEJ[5]. Both NHEJ and MMEJ frequently result in insertions and deletions (indels) of a few nucleotides, which are leveraged for targeted gene disruption[6], but they can also cause larger deletions of several hundred nucleotides or chromosome rearrangements[7,8]. Another end-joining pathway is single-strand annealing (SSA), which requires long stretches (>10 base pairs (bp)) of sequence similarity at both sides of DSBs[9]. Finally, DSBs can be repaired by homology-directed repair (HDR) using sister chromatids as templates, referred to as homologous recombination (HR). HDR using a single-stranded exogenous DNA donor, which can be experimentally provided to the cells, sometimes referred to as single-strand template

repair (SSTR), can be mediated by canonical HR genes, Fanconi anemia genes or DNA mismatch repair genes[10–12].

DNA donors can be designed to introduce nucleotide changes or whole genes into the genome. This holds great promise for treating genetic diseases, as well as for genome-engineering strategies. However, this is difficult because HDR is inefficient compared to NHEJ and MMEJ and because unintended editing events often occur at the targeted genomic sites, as well as elsewhere in the genome[13–16].

Many studies have therefore attempted to increase HDR efficiency by transiently inhibiting proteins central to NHEJ using short interfering (si) RNAs or small molecules[17–20]. For example, we have previously shown that small-molecule inhibition of the DNA-dependent protein kinase catalytic subunit (DNA-PKcs) results in predominant HDR-mediated DSB repair (>50%)[20]. However, this is a finely tuned interaction, as a full *PRKDC* knockout results in proportionally less HDR as it affects the levels of a different kinase (ATM) that is crucial for efficient HDR[21,22]. Others have achieved a modest increase in HDR by inhibiting DNA ligase IV, an enzyme necessary for NHEJ[23].

[1]Department of Evolutionary Genetics, Max Planck Institute for Evolutionary Anthropology, Leipzig, Germany. [2]Human Evolutionary Genomics Unit, Okinawa Institute of Science and Technology, Onna-son, Japan. ✉e-mail: stephan_riesenberg@eva.mpg.de

Like HDR, MMEJ requires resected DSB ends with single-stranded overhangs. The only protein known to have a function exclusive to MMEJ is polymerase theta (Polθ), encoded by the gene *POLQ*. In polymerase theta-mediated end joining, Polθ aligns short nucleotide similarities before DNA synthesis[24,25]. Inhibition of Polθ has been found to slightly increase homologous recombination[26]. However, several other studies show no increase in HDR[27,28]. Other proteins critical for MMEJ are poly(ADP-ribose)-polymerase 1 (PARP1) and DNA ligase I/III, but these are also needed for nick repair and DNA replication. No protein exclusive for SSA has so far been described. For example, RAD52 is central to SSA by promoting annealing of complementary single-stranded DNA, but it also interacts with replication protein A complex (RPA) and RAD51 to stimulate HR[29]. SSTR has been described to be dependent on Rad52 in yeast[12], while RAD52 is dispensable for SSTR in human cells[30]. However, ectopic expression of both RAD52 and a dominant-negative form of tumor protein p53-binding protein 1 improves HDR from single-stranded but not double-stranded DNA donors[30].

To improve the efficiency of precise genome editing by HDR, we introduced mutations in genes necessary for NHEJ, MMEJ and SSA. We found that the combined inhibition of NHEJ by the K3753R mutation in DNA-PKcs and by Polθ V896* (stop codon introduction) in MMEJ results in DSB repair almost exclusively by HDR, while indels, large deletions/rearrangements and off-target editing events are largely abolished. We show that transient inhibition of the two repair pathways using the HDRobust substance mix yields similar results in unmodified human cells.

## Results

### Mutational inhibition of repair genes

To prevent repair of DSBs by NHEJ, MMEJ and SSA while preserving HDR efficiency, we introduced mutations in genes to prevent end joining without affecting HR and thus HDR (Fig. 1a). To inhibit NHEJ, we inactivated the kinase function of DNA-PKcs (K3753R) while keeping other parts of the protein intact. To inhibit MMEJ, we introduced a stop codon (V896*) in *POLQ* that eliminates the DNA polymerase domain and RAD51 binding[31], which may be detrimental for HR by sequestering RAD51. To inhibit SSA, we introduced the three mutations (K152A/R153A/R156A) in RAD52 that abolish DNA binding[32] while keeping RAD51 binding, which stimulates HR, intact. We generated the DNA-PKcs K3753R, Polθ V896* and RAD52 K152A/R153A/R156A (K/R152–156A) mutations singly and in all four possible combinations in H9 human embryonic stem cells (hESCs) carrying an inducible (iCRISPR) Cas9D10A gene[13,19]. Cell population growth was normal for single mutants, reduced for a combination of DNA-PKcs K3753R with Polθ V896* and lowest for combinations containing DNA-PKcs K3753R with RAD52 K/R152–156A (Extended Data Fig. 1)

To test the efficiency of HDR-mediated editing, we performed editing of single positions using single-stranded DNA donors together with transfected gRNA in protein-coding genes *TTLL5*, *RB1CC1*, *VCAN* and *SSH2* (Fig. 1b). Donors were designed such that the mutation of interest also serves as a blocking mutation to prevent recutting. After isolation of DNA, PCR amplicons of the targeted regions were sequenced and HDR was scored as the fraction of amplified molecules that carry the intended nucleotide substitutions. Indels were scored as insertions and deletions with varying lengths of microhomology. When deletions occurred at sites where the sequence on one end of the deletion was identical to the undeleted sequence on the other end and was at least two nucleotides long, we scored this as likely a result of MMEJ, while other indels were attributed to NHEJ[20]. However, some deletions attributed to MMEJ could also be due to NHEJ by chance. Combinations of the targeted nucleotide substitutions and indels were scored as 'imperfect HDR'. We further quantified the ratio of HDR-dependent intended edits to all genome editing events that differ from the wild-type sequence and refer to this as 'outcome purity'.

RAD52 K/R152–156A alone, or its addition to Polθ V896* and/or DNA-PKcs K3753R, did not increase the efficiency of HDR relative to other outcomes (Fig. 1b). Polθ V896* alone did not clearly change HDR efficiency for *VCAN* and *SSH2*, while it increased efficiency for *TTLL5* (21% to 29% HDR) and *RB1CC1* (19% to 41%). DNA-PKcs K3753R alone clearly did not increase HDR efficiency for *VCAN*, slightly increased efficiency for *SSH2* (10% to 16%) and strongly increased efficiency for *TTLL5* (21% to 67%) and *RB1CC1* (19% to 63%). For all targets, deletion patterns of cells carrying repair mutant combinations that include DNA-PKcs K3753R were different from the other cell lines (Extended Data Fig. 2a). The combination of Polθ V896* with DNA-PKcs K3753R strongly increased HDR efficiency for *VCAN* (7% to 33%) and *SSH2* (16% to 37%), and further increased efficiency for *TTLL5* (67% to 80%). Outcome purity was above 91% for all four genes, indicating that inhibition of NHEJ and MMEJ by the combination of Polθ V896* with DNA-PKcs K3753R causes CRISPR-induced DSBs to be repaired almost completely by HDR. This is supported by the observation that this double inhibition reduces mean indels from 82% to 1.7% and results in excessive cell death (at least 95%) when we edited three of the above targets without DNA donors as templates for HDR (Extended Data Fig. 3).

To test how mutations in the DNA repair genes affect the relative amount of HDR in a different cell type and when using CRISPR enzymes that produce different types of DNA breaks, we introduced the DNA-PKcs K3753R, Polθ V896* and RAD52 K/R152–156A mutations singly and in three combinations (no clones could be obtained for the triple mutant combination) in a human myelogenous leukemia line (K562) using the ribonucleoprotein (RNP) high fidelity Cas9 variant (Cas9-HiFi)[33]. In these lines, we edited *TTLL5* with Cas9D10A RNP, *FRMD7* with Cas9-HiFi RNP and *KNL1* with a Cas12a variant (Cpf1-Ultra)[34] RNP. Similar to the results in H9 hESCs and regardless of CRISPR enzyme used, deletion patterns of cells carrying DNA-PKcs K3753R alone or in combination were different from the other cell lines (Extended Data Fig. 2b), and the combination of Polθ V896* with DNA-PKcs K3753R resulted in predominant HDR in the three genes, albeit with reduced efficiency when two different Cas9D10A RNPs were transfected for *TTLL5* double nicking (Fig. 1c). Cas9-HiFi editing of *FRMD7* reached 89% and Cpf1-Ultra editing of *KNL1* reached 78% HDR. Outcome purities ranged from 89 to 97%.

In H9 hESCs as well as K562 cells, inhibition of NHEJ by DNA-PKcs K3753R alone was sufficient to achieve almost complete HDR for the targets *TTLL5*, *RB1CC1* and *FRMD7*, while there was a substantial proportion of deletions with microhomologies for *VCAN*, *SSH2* and *KNL1* (Fig. 1b,c). In line with the assumption that these deletions are due to MMEJ, the combination of Polθ V896* and DNA-PKcs K3753R resulted in virtually no deletions for *VCAN*, *SSH2* and *KNL1*.

We also generated repair mutants of iCRISPR–Cas9D10A 409B2 human induced pluripotent stem cells (hiPSCs) and tested *VCAN*, *SSH2* as well as four other targets for which we had previously observed that DNA-PKcs K3753R alone is not sufficient to achieve outcome purities above 50% (ref. 20). In this case, the combination of DNA-PKcs K3753R with Polθ V896* increased HDR efficiency (4.9-fold) more than DNA-PKcs K3753R alone (3.1-fold) and predominant HDR could be achieved for all targets (mean outcome purity 90%) (Fig. 1d). Editing of 11 additional targets in iCRISPR–Cas9D10A H9 hESC triple repair mutant cells resulted in HDR efficiencies of 21–91% and outcome purities of 77–98% (Fig. 1e). Comparison of outcome purities of all edits in repair gene mutant cells shows that triple mutant cells are not superior to double mutant cells (Fig. 1f).

### Transient repair pathway inhibition

We have recently shown that a small-molecule inhibitor of the active site of DNA-PKcs (M3814; synonyms: nedisertib, peposertib) almost completely blocks NHEJ and transiently increases HDR to an extent comparable to the DNA-PKcs K3753R mutation[20]. To test whether it is possible to also transiently inhibit MMEJ and combine it with NHEJ

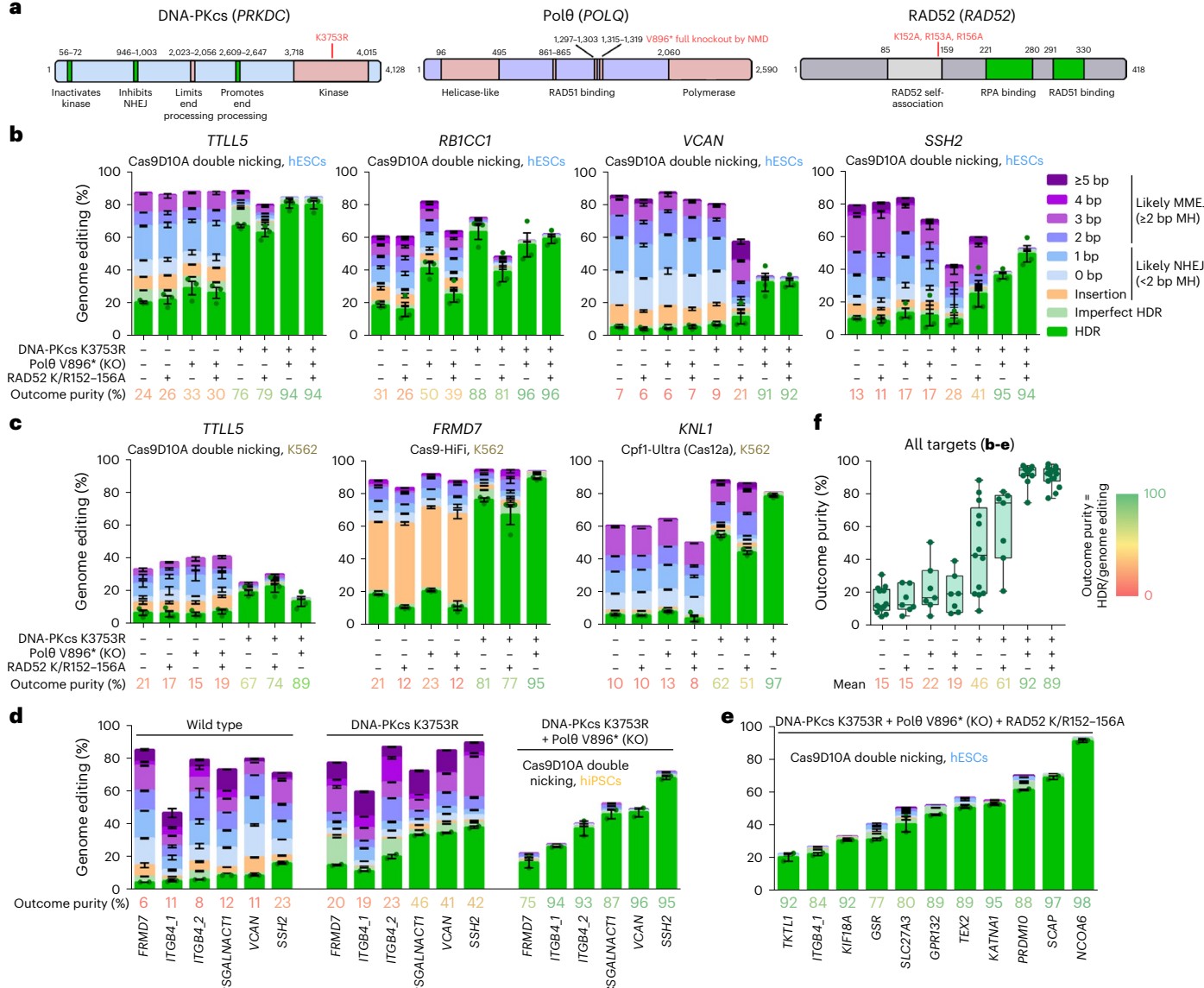

**Fig. 1 | Genome editing efficiencies in cell lines with repair gene mutations.**
**a**, Protein domain structures of DNA-PKcs, Polθ and RAD52. Motifs or domains beneficial or detrimental for HR/HDR are colored green or rose, respectively. The amino acid positions where domains start and end are given and their functions indicated, the positions of mutations are in red. **b**, Genome editing efficiencies using Cas9D10A double nicking in H9 hESCs that carry either no repair gene mutation or combinations of DNA-PKcs K3753R, Polθ V896* and RAD52 K152A/R153A/R156A (K/R152–156A). Frequencies of deletions are presented on the basis of microhomology (MH) length. Independent biological replicates were performed (*n* = 3) and error bars show the s.e.m. For HDR, replicates are depicted by dots. The mean outcome purity given below is the percentage of HDR of all editing events. KO, knockout. **c**, Genome editing efficiencies using Cas9D10A double nicking, Cas9 (HiFi) and Cas12a (Cpf1-Ultra) in K562 cells that carry either

no repair gene mutation or combinations of DNA-PKcs K3753R, Polθ V896* and RAD52 K/R152–156A. Independent biological replicates were performed (*n* = 3) and error bars show the s.e.m. **d**, Genome-editing efficiencies using Cas9D10A double nicking in 409B2 hiPSCs without and with combinations of repair gene mutants of targets for which we have previously shown that DNA-PKcs K3753R alone is not sufficient[20]. Independent biological replicates were performed (*n* = 2) and error bars show the s.e.m. **e**, Genome editing efficiencies of 11 additional targets using Cas9D10A double nicking in H9 hESCs that carry the mutations DNA-PKcs K3753R, Polθ V896* and RAD52 K/R152–156A. Independent biological replicates were performed (*n* = 2) and error bars show the s.e.m. **f**, Outcome purity for all targets in **b–e** for wild-type cells or cells with repair gene mutations. Each dot indicates the mean of one target, boxes the 25th to 75th percentile, lines medians and whiskers extend from minimum to maximum values.

inhibition, we used iCRISPR–Cas9D10A H9 hESCs carrying DNA-PKcs K3753R and a commercial combination of four siRNAs to silence the *POLQ* transcript. When we attempted to edit *VCAN*, for which inhibition of both NHEJ and MMEJ is needed to achieve predominant HDR, outcome purity increased from 38% to 74%, but absolute HDR efficiency did not increase (Extended Data Fig. 4a). However, only one of the four siRNAs used in this experiment targeted sequences upstream of RAD51-binding sites in the *POLQ* transcript[35] (Extended Data Fig. 4b). We speculated that the siRNA-induced cleavage of the mRNA

may result in a truncated protein that may sequester RAD51 and thereby inhibit HDR[26].

To test whether the limitation in HDR efficiency is indeed due to Polθ functions different from its polymerase domain, we generated two mutations (D2540A/E2541A) to eliminate the polymerase activity of Polθ[36] in the cell line while keeping RAD51 binding intact. This drastically increased the outcome purity to 93%, but only slightly increased HDR efficiency to 24% (Extended Data Fig. 4a). Addition of the RAD51 inhibitor B02 (ref. 37) resulted in a dose-dependent decrease

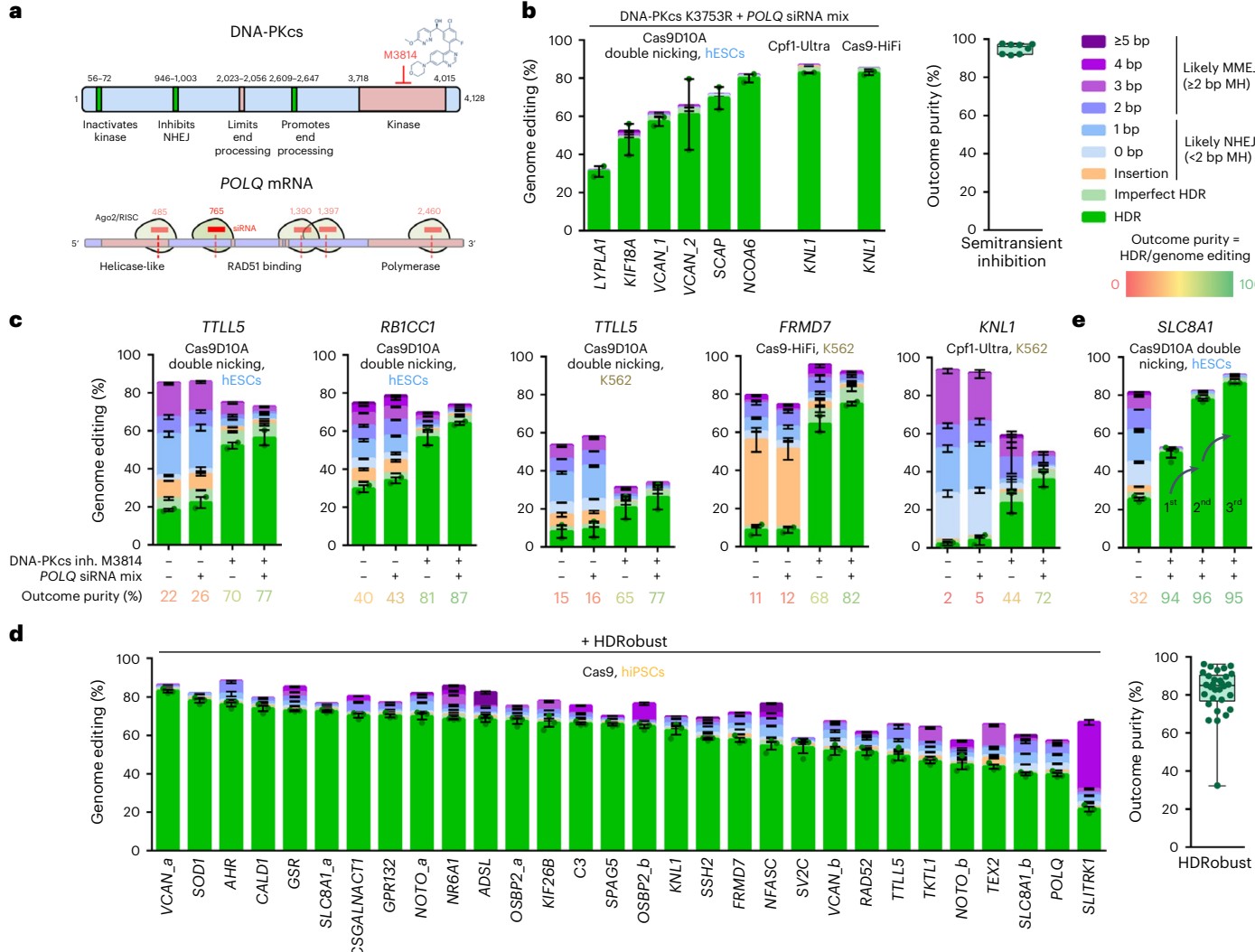

**Fig. 2 | Genome editing using transient inhibition of repair pathways.**
**a**, Strategy for transient repair pathway inhibition. Motifs or domains beneficial or detrimental for homologous recombination/HDR are colored green or rose, respectively. The amino acid positions where domains start and end are given, their functions indicated and the small-molecule inhibitor of the kinase domain of DNA-PKcs M3814 (ref. 76) and siRNAs targeting the *POLQ* mRNA are indicated in red. **b**, Genome editing efficiencies using either iCRISPR–Cas9D10A double nicking, Cas9-HiFi RNP or Cpf1-Ultra RNP (Cas12a) in H9 hESCs that carry DNA-PKcs K3753R and transfected with the *POLQ* siRNAs ('semitransient inhibition', that is, genetic NHEJ inhibition, transient MMEJ inhibition). Frequencies of deletions are presented on the basis of microhomology length. Independent biological replicates were performed (*n* = 2) and error bars show the s.e.m. For HDR, replicates are depicted by dots. The panel to the right gives mean outcome purities (percentage HDR of all editing events) for each target. Each dot indicates the mean of one target, the box the 25th to 75th percentile, lines

medians and whiskers extend from minimum to maximum values. **c**, Genome editing efficiencies using Cas9D10A double nicking in H9 hESCs, as well as using Cas9D10A double nicking, Cas9-HiFi or Cpf1-Ultra in K562 cells and transient inhibition (inh.) of DNA-PKcs by M3814 and/or of *POLQ* by siRNA. Independent biological replicates were performed (*n* = 2) and error bars show the s.e.m. Mean outcome purities are given below the charts. **d**, Genome editing efficiencies of 30 targets using iCRISPR–Cas9 in 409B2 hiPSCs with transient end-joining inhibition by HDRobust (M3814 + *POLQ* siRNA mix). Independent biological replicates were performed (*n* = 3) and error bars show the s.e.m. Each dot in the right panel indicates the mean of one target, the box the 25th to 75th percentile, the line the median and whiskers extend to the minimum and maximum values. **e**, Repeated editing in H9 hESCs with HDRobust increases the percentage of precisely edited cells while maintaining outcome purity. Independent biological replicates were performed (*n* = 3) and error bars show the s.e.m.

of HDR, supporting the importance of RAD51 for HDR in our model system. We therefore tested a custom siRNA targeting exon 15, which is upstream of the RAD51-binding sites. This molecule increased HDR efficiency from 21% to 44% when tested alone. It reduced *POLQ* mRNA levels to 20% in 8 h and was undetectable in later time points until 48 h (Extended Data Fig. 4c). As expected, it was unable to increase HDR in a *POLQ* mutant engineered to express mRNA, with silent codon mutations that prevent binding of this particular siRNA, while amino acids of Polθ are unchanged (Extended Data Fig. 4a). When this siRNA was combined with the siRNA pool (Fig. 2a) the outcome purity after editing

in H9 hESCs carrying DNA-PKcs K3753R for two sites in *VCAN* and six other loci was at least 92%, regardless of whether Cas9-HiFi, Cas9D10A double nicking or Cpf1-Ultra were used (Fig. 2b).

In line with our observation that it is not sufficient to inactivate the polymerase domain of *POLQ* to maximize HDR, small-molecule Polθ polymerase inhibitor ART558 (ref. 38) increased outcome purity to 75%, but not absolute HDR (Extended Data Fig. 5a). Small-molecule Polθ ATPase inhibitor novobiocin[39] had no clear effect (Extended Data Fig. 5b). We also tested two small-molecule inhibitors of PARP and two inhibitors of DNA ligase I/III, two proteins that are involved in MMEJ, but

none of them increased HDR efficiency or outcome purity (Extended Data Fig. 5c–f).

Next, we edited genes previously targeted in the repair gene mutant cells (Fig. 1b) using transient inhibition of NHEJ by M3814 and/or MMEJ by the *POLQ* siRNAs in H9 hESCs and K562 cells expressing wild-type repair proteins. Transient inhibition of only NHEJ increased HDR fivefold, while inhibition of both NHEJ and MMEJ led to a 6.3-fold increase in HDR (Fig. 2c). Outcome purity in the latter case was 72–87%. The combination of M3814 and *POLQ* siRNAs, which we dubbed 'HDRobust', can thus robustly achieve HDR efficiencies comparable to genetic inhibition, albeit with few residual indels for some targets.

We then tested 30 targets with HDRobust using lipofection of gRNA and DNA donors in iCRISPR–Cas9 409B2 hiPSCs. Figure 2d shows that this resulted in predominant HDR for 29 targets (97%) with a mean HDR efficiency across targets of 60% and a mean outcome purity of 82%. When edited without DNA donors, these targets show a wide range of indel signatures, including those that have been described to be unfavorable for HDR due to high MMEJ deletion frequency[40] (Extended Data Fig. 6 and Supplementary Data 1). Editing without DNA donors resulted in 75% to 94% cell survival (mean 87%), while editing using DNA donors and HDRobust resulted in a wider range of 35% to 96% (mean 59%) (Extended Data Fig. 7).

For one target where HDR was 50% after one edit with HDRobust, repeating the editing of the cell population twice with HDRobust increased HDR to 86% (Fig. 2e). When testing repeated cell bulk editing on four additional targets, absolute HDR ranged from 80% to 96%, while outcome purity ranged from 89% to 97% (Extended Data Fig. 8a). Further, cell survival was increased when more cells contain the desired substitution that also serves as a blocking mutation to prevent cleavage (Extended Data Fig. 8b). Thus, cell populations carrying high proportions of precisely edited cells can be produced using HDRobust without the isolation of cellular clones.

## Prevention of unintended on-target effects

In addition to small indels, deletions of a few hundreds or thousands of bases, as well as complex chromosomal rearrangements at the target site, can occur during genome editing[7,8,14,41]. To investigate whether such effects might be prevented by inhibition of NHEJ and MMEJ, we edited two sites in *SCAP* and *TEX2* that we had previously noted were often affected by copy number losses when edited in iCRISPR–Cas9D10A H9 hESCs. We isolated 46–88 cellular clones derived from single cells, sequenced the target site and estimated the copy number of the target sites by droplet digital PCR, to detect deletions, duplications, translocations and chromosome (arm) losses. When *SCAP* and *TEX2* were edited in cells with wild-type repair genes, 8.3% and 13% of the cellular clones were affected by copy number losses, respectively (Fig. 3a). In contrast, when using HDRobust, no cellular clones with losses of target sites were detected among 164 clones analyzed (Fig. 3b). When editing the genes in cells where NHEJ and MMEJ have been inactivated by genetic mutations, similar results were achieved, confirming that the effects are due to the inhibition of the DNA-PKcs kinase function and of Polθ (Fig. 3c–e). Thus, inhibition of NHEJ and MMEJ prevents target copy number loss at the target sites.

## Prevention of off-target effects

To investigate the extent to which inhibition of NHEJ and MMEJ might prevent unintended editing at off-target sites, we introduced nucleotide substitutions in *KATNA1*, *OSBP2* and *RAD52* in H9 hESCs, using gRNAs that in each case matched off-target sites with a single mismatch (Fig. 4a). Three days after editing using Cas9 RNP or Cas9-HiFi RNP with and without HDRobust we scored the editing efficiency at the intended targets as well as at the two most likely predicted off-target sites[42,43] (Fig. 4b). As expected, Cas9-HiFi, which is engineered to reduce off-target editing, reduced the number of off-target deletions from about 60% for the single mismatch off-target sites to 0.3–20%. However,

the outcome purity at the intended targets was 45–68%. When Cas9-HiFi is combined with HDRobust, outcome purity increases to 84–96% and off-target editing is further reduced up to tenfold (0.4–2%) (Fig. 4b). Similar results were achieved in cells where DNA-PKcs and Polθ have been inactivated by mutations (Fig. 4c). Out of the 69 on-target and six off-target sites investigated, we find only one site (*O*-OT-1) where NHEJ and MMEJ inhibition is not sufficient to prevent deletion formation (Extended Data Fig. 9).

After Cas9 editing, double inhibition of NHEJ and MMEJ strongly reduces HDR efficiency at the on-target site, from 34% to 1.3% for *KATNA1*, from 26% to 1% for *RAD52* and from 53% to 20% for *OSBP2* (Fig. 4c). This is contrary to the increases in HDR seen for other targets (Figs. 1–3), which were not selected for having off-target sites that are likely to be cleaved. This suggests that off-target cleavage results in cell death in cells lacking DNA end-joining repair.

In agreement with this, editing of targets prone to off-target editing using Cas9 RNP in unmodified H9 hESCs results in cell survival of 30–50%, while inhibition of both NHEJ and MMEJ by DNA-PKcs K3753R and Polθ V896* (double mutant) results in cell survival of only 10% (Fig. 4d). Using Cas9-HiFi, cell survival is increased to 80% for all targets in unmodified H9 hESCs, and up to 60% or 77% when both NHEJ and MMEJ are blocked by mutations or HDRobust, respectively. Thus, cell survival after editing with double inhibition cells is reduced in cases where off-target cleavage is frequent.

Importantly, the genome stability of proliferating cells is not compromised in double-repair mutant cells compared to wild-type cells after long-term culture with the drug bleomycin, which induces random DSBs (Supplementary Discussion and Supplementary Fig. 1).

## Comparison to prime editing

Prime editing[44] (PE) is currently the method of choice for many applications. It relies on the introduction of single-strand breaks by a Cas9H840A nickase, which is linked to a reverse transcriptase (RT), and that uses the cleaved strand of the target site as a primer to introduce edits from PE gRNA (pegRNA).

To compare the efficiency and outcome purity achieved by HDRobust with PE, we generated a H9 hESC line with an inducible PE system carrying a human codon optimized RT linked to iCRISPR–Cas9H840A (iPrime), and another H9 hESC line where the RT was linked to a Cas9 nuclease variant (iPrimeCut) (Supplementary Data 2). To further improve PE conditions, we used pegRNAs with 5′ and 3′ end phosphorothioate bonds and 2′-OMe residues to prevent degradation of the pegRNA[45]. We tested three targets in the genes *CDKL5* (install c.1412delA), *FANCF* (+5G to T) and *RNF2* (+1C to A) for which the pegRNAs have been optimized[44]. We achieved iPrime PE efficiencies of 1.1% for *CDKL5*, 11.2% for *FANCF* and 18.4% for *RNF2* (Fig. 5a) with outcome purities of 34%, 59% and 78%, respectively. PE at the same positions in other cell types results in varying efficiencies and outcome purities[44,46], but our iPrime PE results are comparable to published efficiencies in hESCs[47] (Fig. 5b). iPrimeCut PE editing efficiencies were slightly higher than with iPrime, but resulted in more indel formation as described[48], and therefore achieved a mean outcome purity of only 21% across the targets. However, when we combined iPrimeCut with HDRobust, PE efficiency increased 3.8-fold and outcome purities were similar to the iPrime results (Fig. 5a).

When electroporating DNA donors and gRNAs that have target sequences identical to the pegRNAs used above, Cas9-HiFi RNP editing with HDRobust achieved HDR efficiencies of 21% for *CDKL5*, 89% for *FANCF* and 91% for *RNF2* with outcome purities of 90–96% (Fig. 5c). Thus, editing with a DNA donor[49] and HDRobust performs better than prime editing (PE3) in terms of absolute precise editing efficiency as well as outcome purity in hESCs, although comparisons to enhanced prime editing methods remain to be performed, and when selecting an editing method, other metrics (for example, viability of a given cell model) may be important to consider as well.

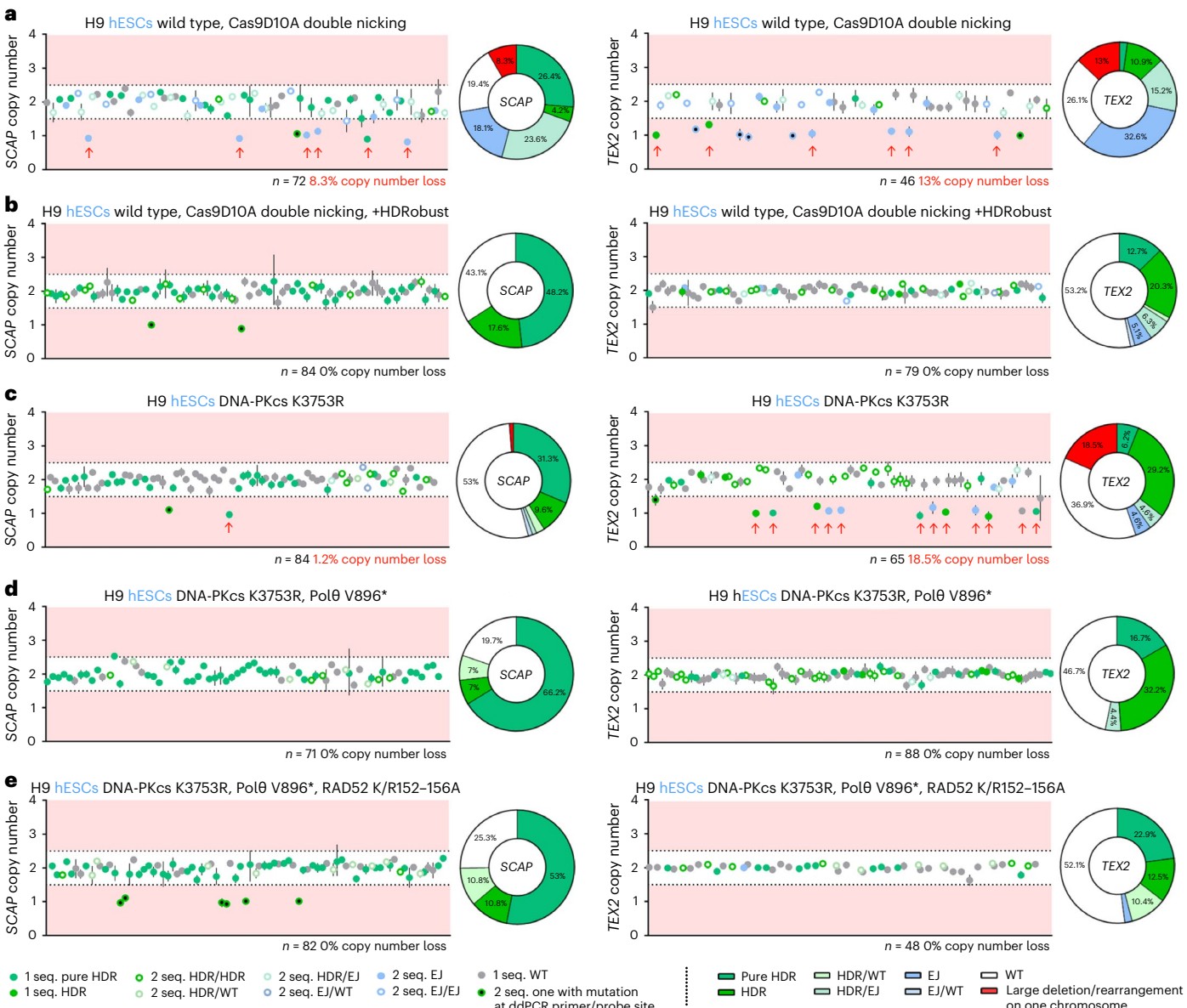

**Fig. 3 | Prevention of unintended on-target effects by HDRobust and genetic end-joining repair inhibition. a–e**, Target site sequencing and droplet digital (dd) PCR copy number analysis of cellular clones after editing of different targets (*SCAP* left panels or *TEX2* right panels) using Cas9D10A double nicking in H9 hESCs without repair gene mutations (**a**), transient end-joining inhibition using HDRobust (M3814 + *POLQ* siRNA mix) (**b**), DNA-PKcs K3753R (**c**), DNA-PKcs K3753R and Polθ V896* (**d**) or DNA-PKcs K3753R, Polθ V896* and RAD52 K152A/R153A/R156A (K/R152–156A) (**e**). The copy number of target sequences relative to the gene *FOXP2* in cellular clones is plotted as a filled or open circle when one predominant DNA sequence (seq.) (apparent homozygous) or two DNA sequences with a similar frequency (apparent heterozygous) were obtained, respectively. The circles are in shades of green and blue to represent different combinations of unmodified chromosomes, chromosomes modified

by HDR and chromosomes modified by NHEJ or MMEJ (summarized as end joining, EJ). Incorporation of the targeted substitution regardless of the presence of additional mutations is quantified as HDR. For cellular clones with one predominant DNA sequence, 'pure HDR' labels the exclusive presence of the targeted substitution. A black dot in a circle fill indicates an indel at the ddPCR primer/probe site that results in inability to amplify this locus for one chromosome. Cellular clones with copy number loss indicative of an on-target effect are labeled with red arrows. The measure of center for the error bars represents the ratio of the Poisson-corrected number of target to reference molecules multiplied by two for the diploid state of the reference gene. The error bars represent the 95% confidence interval of this measurement. The numbers of cellular clones analyzed and percentages of on-target effects are given. Pie charts give the percentage of genotypes of the cellular clones. WT, wild type.

## Correction of disease mutations

To take a step towards investigating the feasibility of ex vivo gene therapy, we used Cas9-HiFi RNP to introduce a nucleotide substitution in *LAG3*, a gene often modified to optimize chimeric antigen receptor-T cells for cancer treatment[50]. While M3814 alone increased HDR efficiency 2.8-fold in primary CD4+ T cells, HDRobust increased HDR efficiency 22.2-fold (0.8 to 16%) and outcome purity from 14 to 51% (Fig. 5d).

We furthermore corrected three different mutations (R459L, R198C and S106C) in the gene encoding glucose-6-phosphate dehydro-genase (G6PD) in lymphoblastoid cells (LCLs) derived from patients suffering from anemia. We also corrected a sickle cell mutation in the hemoglobin gene (*HBB* E6V) and a mutation in the prothrom-bin gene resulting in thrombophilia (*F2* c.*97G>A also known as c.20210G>A). HDRobust increased mean HDR efficiency across the

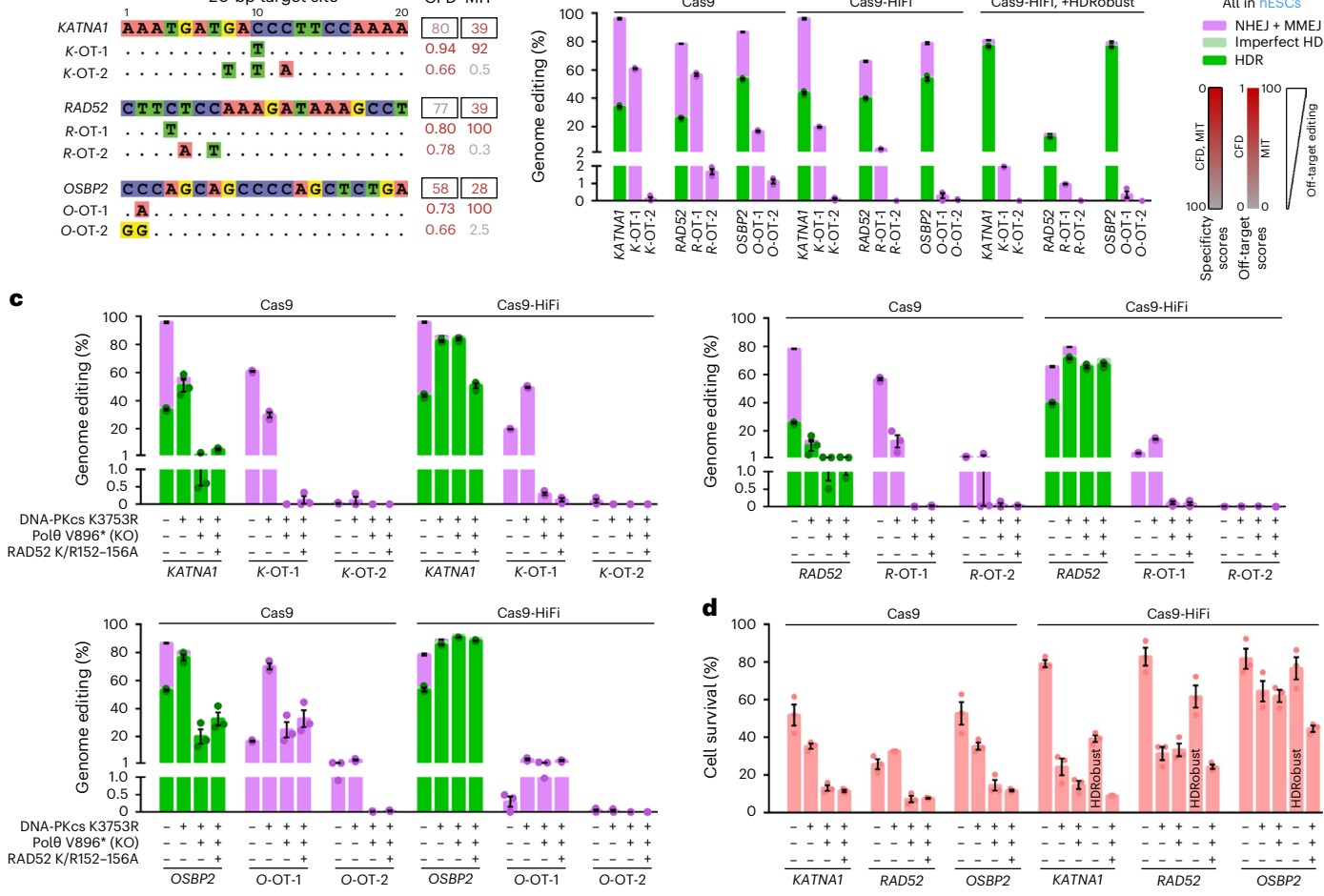

**Fig. 4 | Prevention of off-target effects by HDRobust and genetic end-joining repair inhibition. a**, Target site sequences of three different gRNAs that are predicted to be prone to off-target editing using CFD[42] and MIT[77] specificity scores. The sequences of the two off-targets with the highest CFD scores are shown below the on-target sites. Identical bases are given by dots. The CFD and MIT scores are in black frames. **b**, Genome editing efficiencies with Cas9, Cas9-HiFi or Cas9-HiFi with HDRobust at the on-target and top two CFD off-target sites in H9 hESCs without repair gene mutations. Independent biological replicates were performed (*n* = 3) and error bars show the s.e.m. **c**, Genome editing efficiencies at the on-target and top two CFD off-target sites by Cas9 RNP or

Cas9-HiFi RNP in H9 hESCs without repair gene mutations, as well as with combinations of DNA-PKcs K3753R, Polθ V896* and RAD52 K152A/R153A/R156A (K/R152–156A). Independent biological replicates were performed (*n* = 3) and error bars show the s.e.m. **d**, Cell survival after editing with Cas9 RNP or Cas9-HiFi RNP in H9 hESCs without repair gene mutations, in combinations of DNA-PKcs K3753R, Polθ V896* and RAD52 K/R152–156A, as well as with HDRobust and Cas9-HiFi in wild-type cells. Cell survival was quantified by a fluorescence resazurin assay with respect to mock electroporation without editing. Independent biological replicates were performed (*n* = 3) and error bars show the s.e.m. Replicates are depicted by dots for HDR, NHEJ + MMEJ, or cell survival.

---

five targets from 4–34% to 17–72% and increased outcome purity from 32–51% to 67–90% (Fig. 5e).

### Generation of brain organoids

To assess whether editing in conjunction with inhibition of NHEJ and MMEJ might negatively influence the ability of stem cells to differentiate, we edited a nucleotide in the gene *NOVA1* to change a valine at position 197 in the encoded protein to an isoleucine seen in Neandertals and non-human primates. In in vitro three-dimensional cultures, this change has been reported to result in bumpier and smaller brain organoids during the proliferation stage than those derived from unedited stem cells[51].

We used the gRNA target, DNA donor and Cas9 as in the published work[51], as well as Cas9-HiFi RNP and HDRobust, to achieve efficient HDR efficiency and prevent unintended on-target and off-target effects, to edit 409B2 hiPSCs. Cas9-HiFi and HDRobust increased HDR efficiency from 34% to 83% and reduced the percentage of cellular clones with aberrant *NOVA1* copy number from 69% to 3% (Fig. 6a–d). We also amplified and sequenced heterozygous single-nucleotide polymorphisms upstream and downstream of the target site to check for loss of

heterozygosity that occurs when sister chromatids are used to repair DSBs (Fig. 6e). Subsequent organoid differentiation worked equally well in the wild type and in the edited cells suggesting that HDRobust does not affect the ability of the cells to differentiate to organoids. Organoid shape and size were not affected by the ancestral *NOVA1* mutation (Fig. 6f,g and Extended Data Fig. 10), compatible with the conclusion that the morphological and other organoid alterations previously observed[51] might be due to unintended on-targets effects[52,53]. In conclusion, our results demonstrate that genome editing in conjunction with HDRobust does not negatively influence the ability of 409B2 hiPSCs to differentiate into brain organoids.

### Discussion

Methods that enable the introduction of precise changes in genomes can be powerful tools in our genome editing arsenal for both research and clinical applications. For example, the introduction of deletions to inactivate genes is now implemented in many organisms to study the roles of specific genes[54]. In medicine, it holds great promise as a potential treatment for genetic diseases.

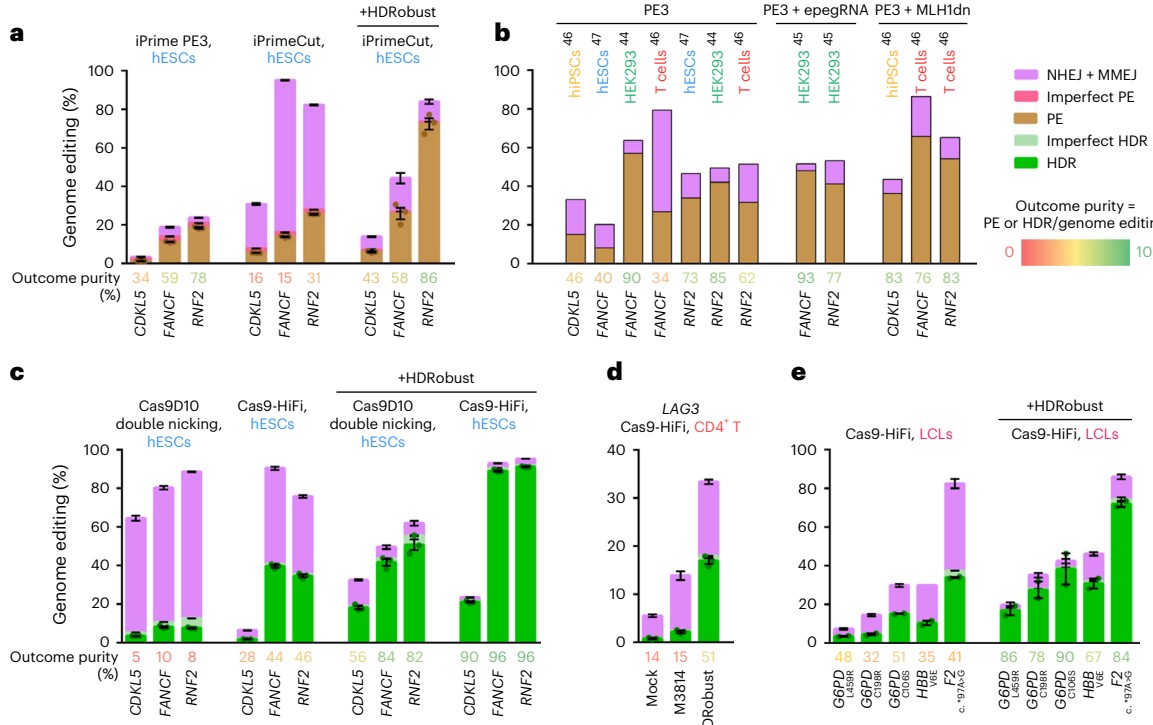

**Fig. 5 | Prime editing with HDRobust, comparison with standard editing and HDRobust and correction of disease mutations. a**, Genome editing efficiencies using iPrime (Cas9H840A nickase), iPrimeCut (Cas9 nuclease prime editor) or iPrimeCut with HDRobust (M3814 + *POLQ* siRNA mix) in H9 hESCs that carry no repair gene mutation. Independent biological replicates were performed (*n* = 3) and error bars show the s.e.m. For PE, replicates are depicted by dots. The mean outcome purities (percentage PE of all editing events) are given. **b**, Published editing efficiencies of mutations at the same positions for *CDKL5*, *FANCF* and *RNF2* in hiPSCs, hESCs, HEK293 or T cells[44–47] are shown. **c**, Genome editing efficiencies of the same sites as in **a** with iCRISPR–Cas9D10A double

nicking, or Cas9-HiFi RNP with or without HDRobust in H9 hESCs that carry no repair gene mutation. Independent biological replicates were performed (*n* = 3) and error bars show the s.e.m. For HDR, replicates are depicted by dots. Mean outcome purities (percentage HDR of all editing events) are given. **d**, Genome editing efficiencies using Cas9-HiFi in primary CD4⁺ T cells with M3814 alone or HDRobust. Independent biological replicates were performed (*n* = 3) and error bars show the s.e.m. **e**, Genome editing efficiencies using Cas9-HiFi in patient-derived LCLs with or without HDRobust. Independent biological replicates were performed (*n* = 2) and error bars show the s.e.m.

However, precise single-nucleotide genome editing has so far been limited by two technical problems. First, DNA breaks introduced by CRISPR–Cas9 need to be repaired by HDR, which allows nucleotide substitutions to be introduced from a template provided to the cells. However, NHEJ and MMEJ, which tend to introduce deletions, occur much more frequently than HDR. While this has been partly overcome by the development of base editors[55,56], which convert one base to another by deamination, without introduction of DNA DSBs, it can only be applied to cytosines and adenines. Base editing efficiency is also limited and currently only 46% of pathogenic T>C variants and 34% of pathogenic G>A variants can be precisely edited by C>T base editors and A>G base editors, respectively[57]. Second, unintended genomic changes at the target site, as well as other sites in the genome that have sequence similarity to the intended targets, often occur. This can create problems within experimental systems and are a serious concern for therapeutic applications.

Here, we have developed HDRobust to overcome these challenges. We have shown that inhibition of DNA-PKcs kinase activity (needed for NHEJ) and Polθ (needed for MMEJ), either by nucleotide substitutions or by small molecules and siRNA, strongly improves single-nucleotide editing by HDR.

It may appear surprising that simultaneous delivery of siRNA with CRISPR reagents is sufficient to quantitatively inhibit MMEJ of CRISPR-induced DSBs. Repair of DSBs often lasts for more than 20 h in mammalian cells, and MMEJ shows delayed activity compared to other repair pathways[58,59]. Therefore, the high efficacy of the *POLQ* siRNA mix

to prevent MMEJ is probably due to fast and long-lasting *POLQ* mRNA knockdown, fast turnover of Polθ protein[60], and strong and persistent binding of the Cas9–gRNA complex to its target, for hours[49,61], before DSBs are accessible to DNA repair.

Our observations that *POLQ* siRNA results in higher absolute HDR than small-molecule inhibition of the polymerase function of Polθ, and that small-molecule inhibition of RAD51 decreases HDR, is compatible with the hypothesis that RAD51-binding sites in Polθ limit HDR[26]. Notably, we and others have identified RAD51 to be required for HDR when single-stranded DNA donors are provided[10,19,20], while several studies find RAD51 to be dispensable for SSTR[11,12,62,63]. This might be due to cell type-specific differential reliance on repair subpathways, or initial RAD51-independent SSTR of one chromosome and subsequent RAD51-dependent HR utilizing the already repaired chromosome as template. Further studies will be needed to clarify this.

Although HDRobust is a powerful tool, there remain some limitations. In some rare cases, as we show for one off-target site, NHEJ and MMEJ inhibition are not sufficient to prevent deletion formation. The reason is likely to be that in this case both sides of the cut have long stretches of sequence similarity, resulting in ≥16 bp sequences on one end of deletions that are identical to the undeleted sequences. Such long stretches of sequence similarity may allow annealing of the DNA strands without the help of repair proteins, since even inhibition of SSA in addition to NHEJ and MMEJ is unable to reduce deletions at this site. Another limitation to HDRobust is that HR and thus HDR are restricted to dividing cells. However, activation of HR in G1 phase of the cell cycle

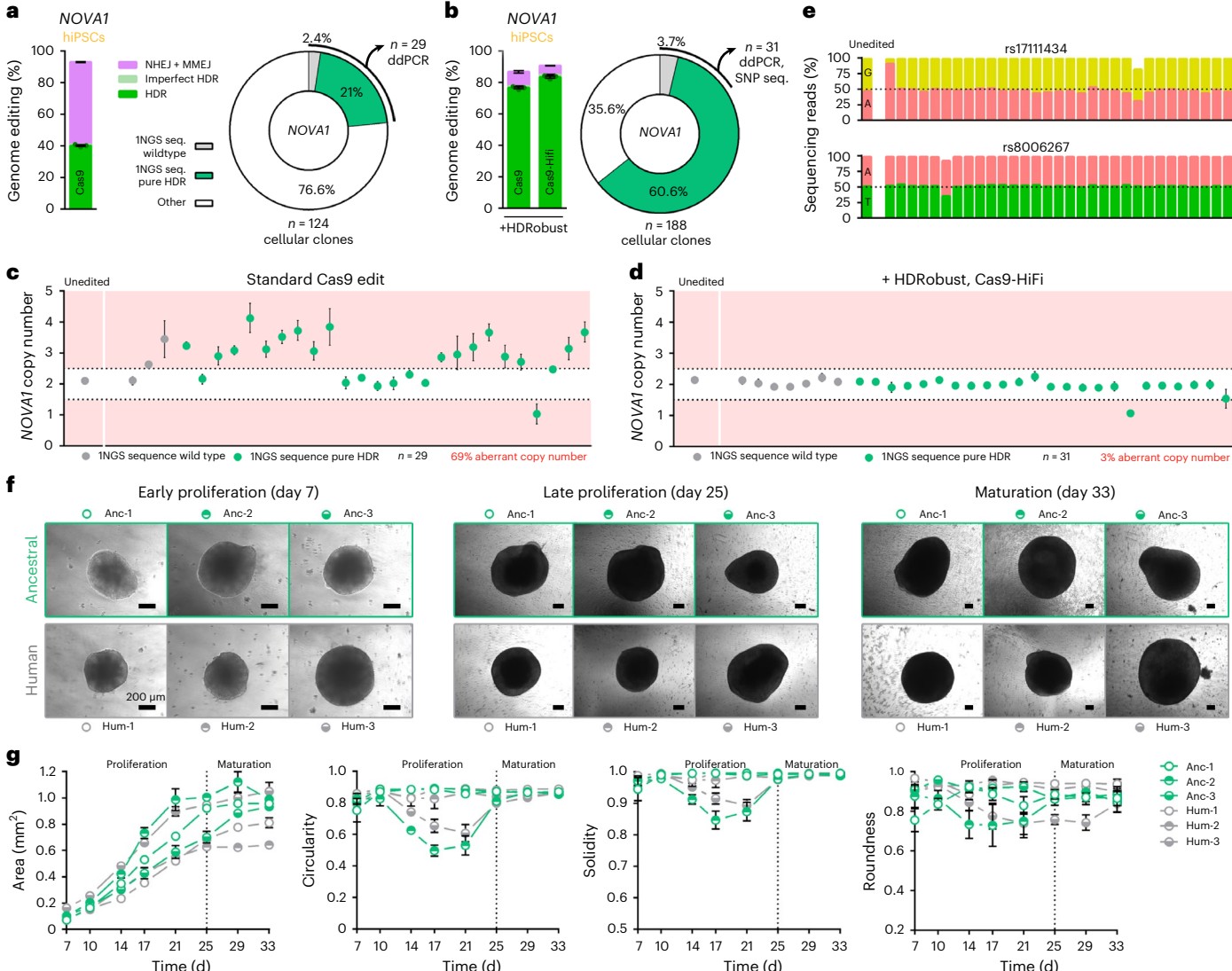

**Fig. 6 | Brain organoid morphology after editing of *NOVA1* to the Neandertal state. a**, Standard genome editing of *NOVA1* as described in[51]. Independent biological replicates were performed (*n* = 3) and error bars show the s.e.m. For HDR, replicates are depicted by dots. The right panel shows the percentage of single cell-derived colonies that have one predominant wild-type DNA sequence (apparent homozygous wild type, gray), one predominant 'pure' HDR sequence (apparent homozygous exclusive ancestral edit, green) or any other genotype (white). The total number of analyzed cellular clones is given. **b**, Genome editing efficiencies and genotypes of single cell-derived colonies after editing of *NOVA1* as done in **a**, but with HDRobust and Cas9-HiFi RNP. Independent biological replicates were performed (*n* = 3) and error bars show the s.e.m. **c**, Copy number (relative to the *FOXP2* gene) of the target site of the single cell-derived cellular clones from **a** that appear homozygous at the target site for wild type and 'pure' HDR on the basis of sequencing of the target site. Copy number estimates are plotted as gray and green circles for wild type and 'pure' HDR clones, respectively. The measure of center for the error bars represents the ratio of the Poisson-corrected number of target to reference molecules multiplied by two for the diploid state of the reference gene. The error bars

represent the 95% confidence interval of this measurement. The number of analyzed cellular clones and percentage of clones with aberrant copy numbers of the target site are given. **d**, Copy number of the target site of the cellular clones from **b** that appear homozygous at the target site for wild type and 'pure' HDR on the basis of sequencing of the target site. The measure of center for the error bars represents the ratio of the Poisson-corrected number of target to reference molecules multiplied by two for the diploid state of the reference gene. The error bars represent the 95% confidence interval of this measurement. **e**, Genotypes of SNPs upstream (rs17111434) and downstream (rs8006267) of the target site from the cellular clones in **d**. **f**, Phase-contrast images of three typical cellular clones with the modern human (Hum-1–3, gray circles) or ancestral (Anc-1–3, green circles) *NOVA1* during early proliferation (day 7), late proliferation (day 25) and maturation (day 33). **g**, Organoid size and shape descriptors of circularity, solidity and roundness during brain organoid development from proliferation to maturation (days 7–33). Data for three different cellular clones for human (gray circles) and ancestral (green circles) are given. Circles show the mean and error bars show the s.e.m. of measurements of four different organoids for each day and clone.

may be possible[64] and hESCs and hiPSCs can obviously be differentiated into nondividing cell types of interest after editing[65]. Finally, inhibition of NHEJ and MMEJ is suitable for cells, but not for editing in organisms, except perhaps in some animal models.

Nevertheless, the precision of HDRobust opens a plethora of opportunities. We have shown that HDRobust performs well with

different CRISPR enzymes (Cas9, Cas9D10A, Cas12a/Cpf1), cell types (hESCs, hiPSCs, K562 cells, primary CD4+ T cells, LCLs) and modes of delivery (electroporation and lipofection). It is also encouraging that the percentage of edited cells can be increased by two or three consecutive rounds of edits without any increase in the frequency of deletions. This opens the possibility of generating populations of cells where the

majority are edited without the need to generate cellular clones from single cells, not only reducing work load, but also preventing clone-to-clone variation, which often complicates analyses of edited cells[66].

Next, similar to prime editing[44], HDRobust can introduce all 12 types of point mutations, as well as insertions and deletions, when provided with a suitable DNA donor. Thus, it has the potential to correct 89% of the pathogenic variants associated with human diseases in the ClinVar database[44,67]. Although not tested here, HDRobust is also likely to increase outcome purity and further increase already high efficiencies achieved when using AAV6 donors, which hold great promise for therapeutic gene editing[33,68–70]. However, HDRobust has not yet been validated for clinical development.

Because dual inhibition of NHEJ and MMEJ prevents copy number loss and off-target editing, we speculate that cells lacking these end-joining pathways cannot repair DSBs in the absence of a suitable DNA donor, resulting in the fact that they can only repair breaks using the exogenous DNA donor or sister chromatids at the on-target site, and solely sister chromatids at off-target sites. When DSBs cannot be repaired in one of these ways, especially when excessive off-target cleavage occurs, cells will die, resulting in a population of precisely edited and wild-type cells. Thus, both unintended indels and large-scale modifications associated with CRISPR cleavage (large deletions, duplications, inversions, translocations, chromothripsis[7,41,71]) can be prevented. This is supported by the observation that dual loss of Polθ and ligase IV abolishes integration of exogenous DNA in human cells[72,73]. Interestingly, double inhibition could potentially be used to screen for low-specificity gRNAs without actually having to identify the potential off-target sites.

Finally, HDRobust can be extended to cells from many other species, as the lysine residue at position 3753 in DNA-PKcs is conserved among vertebrates, and DNA-PKcs itself is widely distributed among invertebrates, fungi, plants and protists[74]. Similarly, homologs of Polθ exist in all or most multicellular eukaryotes[75], making HDRobust widely applicable as a genome editing approach.

## Online content

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

## Methods

### Cell culture

Stem cell lines used were: hESCs (WiCell Research Institute, catalog no. WA09, ethics permit AZ 3.04.02/0118), modified H9 hESCs carrying iCRISPR–Cas9D10A[13,78] and 409B2 hiPSC (Riken BioResource Center, catalog no. HPS0076, GMO permit AZ 54-8452/26) carrying either iCRISPR–Cas9 or iCRISPR–Cas9D10A[19]. We modified iCRISPR H9 hESCs to carry a reverse transcriptase adjacent to Cas9H840A[44] (iPrime) or Cas9 (ref. [48]) (iPrimeCut) (Supplementary Data 2). Stem cells were grown on Matrigel Matrix (Corning, catalog no. 35248) in mTeSR1 medium (StemCell Technologies, catalog no. 05851) with supplement (StemCell Technologies, catalog no. 05852) that was replaced daily. At ~80% confluence, stem cells were dissociated using EDTA (VWR, catalog no. 437012C) and split 1:6 to 1:10 in medium supplemented with 10 μM Rho-associated protein kinase (ROCK) inhibitor Y-27632 (Calbiochem, catalog no. 688000) for one day after replating. To generate cellular clones, H9 hESCs were treated with TrypLE (Gibco, catalog no. 12605010) for 5 min at 37 °C and triturated before seeding 1:100 to 1:500 in mTeSR1 containing (ROCK) inhibitor Y-27632. After at least 7 d, colonies were picked.

Human immortalized myelogenous leukemia cells (K562) (ECACC, catalog no. 89121407) were grown in Iscove's modified Dulbecco's media (ThermoFisher, catalog no. 12440053) with 10% FBS. CD4+ T cells (HemaCare, catalog no. PB04C-1) were grown in RPMI 1640 (ThermoFisher, catalog no. 11875-093) with 10% FBS and activated with Dynabeads Human T-Activator (CD3/CD28) (ThermoFisher, catalog no. 11131D). LCLs (Coriell Institute, catalog nos. GM14890, HG02367, GM16265, GM08369) were cultured in RPMI 1640 with 15% FBS at 37 °C in a humidified incubator with 5% CO$_2$. Media was replaced every second day and cells were split 1:6 to 1:10 once per week.

All cell lines were authenticated by the supplier via certificate of analysis and additionally in-house by checking morphology. All cell lines were tested negative for mycoplasma contamination before and after the experiments.

### Small molecules and oligonucleotides

Commercially available small molecules used were: M3814 (MedChemExpress, catalog no. HY-101570), Rucaparib (MedChemExpress, catalog no. HY-10617A), AG-14361 (MedChemExpress, catalog no. HY-12032), L67 (MedChemExpress, catalog no. HY-15586), L189 (MedChemExpress, catalog no. HY-15588), ART558 (MedChemExpress, catalog no. HY-141520), novobiocin (MedChemExpress, catalog no. HY-B0425), B02 (Sigma, catalog no. SML0364) and bleomycin (Sigma, catalog no. B8416). All gRNAs, DNA donors and primers were from Integrated DNA Technologies (Supplementary Data 2). SiRNAs were the predesigned smart pool-containing siRNAs 485, 1390, 1397 and 2460 (Horizon Discovery, ON-TARGET plus Human POLQ siRNA - SMART 10721) and the siRNA 765 (Integrated DNA Technologies, DsiRNA hs.Ri.POLQ.13.8).

### Electroporation

Adherent stem cells as well as LCLs with a tendency to clump were treated with TrypLE (Gibco, catalog no. 12605010) for 5 min at 37 °C and triturated to obtain single cells, before addition of preheated media. Cells were counted using the Countess Automated Cell Counter (Invitrogen) and cell suspensions were centrifuged at 300g for 3 min at room temperature. iCRISPR cells were incubated in medium containing 2 μg ml⁻¹ doxycycline (Clontech, catalog no. 631311) 3 d before editing to express Cas9, Cas9D10A or the prime editors, respectively. For stem cells without integrated iCRISPR, we used recombinant *Streptococcus pyogenes* Cas9, Cas9-HiFi (R691A) and Cas9D10A proteins, as well as the *Acidaminococcus* sp. BV3L6 Cas12a (Cpf1-Ultra) protein from Integrated DNA Technologies. Electroporation for all cell types (except LCLs) was done using the B-16 program of the Nucleofector 2b device (Lonza) in cuvettes for 100 μl Human Stem Cell nucleofection buffer (Lonza, catalog no. VVPH-5022), containing 1 million cells, 100 pmol

electroporation enhancer, 320 pmol gRNA (crRNA/tracR duplex for Cas9 and its variants and crRNA for Cas12a) (or 640 pmol pegRNA and 214 pmol nicking gRNA for prime editing) and 200 pmol of each single-stranded DNA donor. Where applicable, we added 252 pmol CRISPR enzyme, 160 pmol of *POLQ* siRNA predesigned pool and 320 pmol of *POLQ* siRNA 765.

For LCLs, electroporation of LCLs was done using the T-020 program[79] of the Nucleofector 2b Device (Lonza) in cuvettes for 100 μl Kit V buffer (Lonza, catalog no. VCA-1003) containing 4 million cells, 100 pmol electroporation enhancer, 640 pmol of gRNA (crRNA/tracR duplex), 450 pmol of each single-stranded DNA donor and 252 pmol CRISPR–Cas9-HiFi. Where applicable, we added 320 pmol of *POLQ* siRNA predesigned pool, and 640 pmol of *POLQ* siRNA 765. For transient NHEJ inhibition, 2 μM M3814 was added for 2 d after electroporation.

### Lipofection

409B2 iCRISPR–Cas9 hiPSCs were incubated in medium containing 2 μg ml⁻¹ doxycycline (Clontech, catalog no. 631311) 3 d before lipofection to express Cas9. Lipofection was done with a final concentration of 15 nM of gRNA (crRNA/tracR duplex), 7.5 nM of *POLQ* siRNA predesigned pool, 15 nM of *POLQ* siRNA 765 and 10 nM of single-stranded DNA donor. In brief, 0.75 μl RNAiMAX (Invitrogen, catalog no. 13778075) and the respective oligonucleotides were separately diluted in 25 μl OPTI-MEM (Gibco, catalog no. 1985-062) and incubated at room temperature for 5 min. Both the RNAiMAX and the oligonucleotide dilutions were mixed to yield 50 μl of OPTI-MEM including RNAiMAX, gRNAs and single-stranded DNA donor. The lipofection mix was incubated for 20–30 min at room temperature. Cells were dissociated using EDTA for 5 min and counted using the Countess Automated Cell Counter (Invitrogen). The lipofection mix, 100 μl containing 25,000 dissociated cells in mTeSR1 supplemented with Y-27632, 2 μg ml⁻¹ doxycycline and 2 μM M3814 were put in one well of a 96-well plate covered with Matrigel Matrix (Corning, catalog no. 35248). After 24 h, the medium was replaced with mTeSR1 containing 2 μM M3814 and after one additional day with mTesR1 without M3814.

### Illumina library preparation and sequencing

Five days or more after transfection cells were dissociated using TrypLE (Gibco, catalog no. 12605010), pelleted and resuspended in 15 μl QuickExtract DNA extraction solution (Lucigen, catalog no. QE09050). Incubation at 65 °C for 10 min, 68 °C for 5 min and finally 98 °C for 5 min was performed to yield single-stranded DNA. PCR was done in a T100 Thermal Cycler (Bio-Rad) using the KAPA2G Robust PCR Kit (Sigma, catalog no. KK5024) with supplied buffer B and 3 μl of cell extract in a total volume of 25 μl. The thermal cycling was: 95 °C 3 min; 34× (95 °C 15 s, 65 °C 15 s, 72 °C 15 s); 72 °C 60 s. Illumina adapters (P5 and P7) with sample-specific indices were added in a second PCR reaction[80] using Phusion HF MasterMix (Thermo Scientific, catalog no. F-531L), 0.3 μl of the first PCR product and cycling was: 98 °C 30 s; 25× (98 °C 10 s, 58 °C, 10 s, 72 °C 20 s); 72 °C 5 min. Amplifications were analyzed using 2% EX agarose gels (Invitrogen, catalog no. G4010–11), indexed amplicons were purified using solid phase reversible immobilization beads in a 1:1 ratio of beads to PCR solution[81]. Double-indexed libraries were sequenced on a MiSeq (Illumina) giving paired-end sequences of 2 × 150 bp (+7 bp index). After base calling using Bustard (Illumina), adapters were trimmed using leeHom[82].

### Amplicon sequence analysis

Bam files were demultiplexed and converted into fastq files using SAMtools[83]. Fastq files were used as input for CRISPResso[84] to analyze sequencing read percentage of wild type (unedited), targeted nucleotide substitution (HDR or PE in case of prime editing), indels (NHEJ and MMEJ) and mix of both (imperfect HDR or imperfect PE in case of prime editing). Analysis was restricted to amplicons with a minimum of 70%

similarity to the wild-type sequence and to a window of 20 bp from each gRNA. Sequence similarity for an HDR occurrence was set to 95%. Unexpected substitutions were ignored as putative sequencing errors. We further employed a Python script to identify sequencing reads with indels to be a likely a result of NHEJ (<2 bp microhomology at deletion) or MMEJ (≥2 bp microhomology at deletion)[20]. Sequencing data from single cell-derived cellular clones was analyzed using SAMtools.

### Droplet digital and quantitative PCR

Copy numbers of target sequences were estimated by quantitative ddPCR. Primers were designed flanking the cut site and the probe was designed excluding edited sites. The gene *FOXP2* was used as copy number reference. The ddPCR amplification was done in 1× ddPCR Supermix for probes (no dUTP, Bio-Rad, catalog no. 1863024), 0.2 µM primer and 0.2 µM probe for target and reference, together with 1 µl genomic DNA in QuickExtract DNA extraction solution (Lucigen, catalog no. QE09050). After droplet generation, the PCR reaction for *SCAP*, *TEX2*/*NOVA1* was run for 5/10 min at 95 °C, followed by 42/40 cycles of 35/30 s at 95 °C (at a ramp rate of 1.5/2 °C s$^{-1}$) and 65/60 s at 60/59 °C (at a ramp rate of 1.5/2 °C s$^{-1}$) and 5 min at 98 °C. Droplets were read in a QX200 Droplet reader (Bio-Rad) and allele copy numbers were determined relative to a different fluorophore for the *FOXP2* reference and unedited control.

For the siRNA knockdown time course, RNA was extracted using ice-cold QuickExtract RNA extraction solution (Lucigen, catalog no. QER09015) and reverse transcribed into cDNA with the High Capacity cDNA Reverse Transcription kit (Applied Biosystems, catalog no. 4368814) and the thermal profile: 25 °C 10 min, 50 °C 30 min, 85 °C 5 min. Quantitative PCR (qPCR) was done using CFX96 Real-Time-System C1000 Touch (Bio-Rad) and the Maxima SYBR Green qPCR Master mix no ROX (Thermo Scientific, catalog no. K0253). The thermal profile of the qPCR was: 95 °C 10 min, 45× (95 °C 30 s, 60 °C 30 s, 72 °C 30 s) (for primers see Supplementary Data 2).

### Resazurin assay

Subsequent to editing, cells were grown in media containing ROCK inhibitor Y-27632 for 1 d, followed by normal media for 2 d before being supplied with fresh media containing 10% resazurin solution (Cell Signaling, catalog no. 11884) and grown for 5 h before fluorescence readings using a Typhoon 9410 imager (Amershamn Biosciences) and quantification using ImageJ and the 'ReadPlate' plugin (Fig. 4 and Extended Data Fig. 7), or grown for 2 h before fluorescence readings using a CLARIOstar imager (BMG Labtech) (Extended Data Figs. 1, 3 and 8). Resazurin is converted into fluorescent resorufin by cellular dehydrogenases and fluorescence (excitation: 530–570 nm, emission: 590–620 nm) reflects the amount of living cells[85]. Wells with media and resazurin but without cells were used as blank.

### Brain organoids

We generated cortical organoids as previously described[86] with minor changes. In brief, cells were detached using Accutase (Sigma, catalog no. A6964) for 3 min at 37 °C and 9,000 cells per clonal cell line were seeded in low-attachment 96-well plates (Corning) in 150 µl mTeSR1 media (StemCell Technologies, catalog no. 05851) with supplement (StemCell Technologies, catalog no. 05852) and 10 µM ROCK inhibitor (Calbiochem, catalog no. 688000) for the first two days after seeding. The plates were centrifuged 3 h after seeding at 200 *g* for 1 min to concentrate the cells in the middle of the well. Starting 48 h past seeding, until day 5, the initial media was diluted out with human pluripotent stem (hPS) cell media by carefully aspirating 100 µl and adding 100 µl of fresh hPS cell media. hPS cell media consisted of DMEM/F12, knockout serum 20%, GlutaMax 1:200, nonessential amino acids 1:100, Pen-Strep 1:100 (all Life Technologies), and 2-mercaptoethanol 100 µM (Sigma, catalog no. M3148) supplemented with 10 µM SB-431542 (Abcam, catalog no. ab120163) and 5 µM dorsomorphin (Sigma, catalog no. P5499).

On day 6, the medium was changed to neural medium (NM) consisting of Neurobasal A (Life Technologies, catalog no. 10888-022), B27 supplement (no vitamin A) (Life Technologies, catalog no. 12587010) and GlutaMax 1:100 (Life Technologies, catalog. no. 35050-061) supplemented with 20 ng ml$^{-1}$ EGF (Millipore, catalog no. 01-102) and 20 ng ml$^{-1}$ FGF2 (R&D Systems, catalog no. 233-FB). Organoids were cultured in this medium for the next 19 d with daily medium changes in the first 10 d and every second day for the remaining 9 d. From day 25 on, FGF2 and EGF2 were replaced with 20 ng ml$^{-1}$ BDNF (PeproTech, catalog no. 450-02) and 20 ng ml$^{-1}$ NT3 (PeproTech, catalog no. 450-03) in the NM with medium changes every second day. Starting at day 43, only NM without any growth factors was used with medium changes every second day. We acquired a time course of phase-contrast images of organoids, extracted the two-dimensional (2D) shapes using the polygonal lasso tool of Adobe Photoshop CS5 software and quantified 2D shape descriptors of each organoid using ImageJ.

### Karyotyping

H9 hESCs that carry no repair gene mutation (wild type) or both DNA-PKcs K3753R and Polθ V896* (double mutant) were treated with different tenfold-diluted concentrations (1 µg ml$^{-1}$ to 0.1 ng ml$^{-1}$) of bleomycin to determine the highest concentration that still allows propagation of both cell lines. Both cell lines were then propagated in media containing 1 ng ml$^{-1}$ bleomycin for 5 months before karyotyping. Trypsin-induced Giemsa staining (GTG) or spectral karyotyping (SKY) were carried out according to international quality guidelines (ISCN 2016: An International System for Human Cytogenetic Nomenclature[87] by the 'Sächsischer Inkubator für klinische' (Leipzig).

### Statistics and reproducibility

Bar graphs in figures were plotted and s.e.m. error bars were calculated using GraphPad Prism 6 software. The number of replicates is stated in the respective figure legends. No statistical method was used to predetermine sample size. The experiments were not randomized. Samples were prepared unblinded but in parallel. Analysis was performed on the basis of numerical sample names, without the identity of the samples being known during the analysis.

### Reporting summary

Further information on research design is available in the Nature Portfolio Reporting Summary linked to this article.

## Data availability

The sequencing data generated in this study have been deposited in the Dryad database under accession code dryad.fj6q5740f. Source data are provided with this paper. Data are also available on request from the authors.

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

## Acknowledgements

We thank A. Weihmann and B. Schellbach for DNA sequencing and H. Holland for karyotyping. Funding was provided by the Max Planck Society (S.P.) and the NOMIS foundation (S.P.).

## Author contributions

S.R. conceived the idea. S.R., P.K. and D.M. performed editing experiments and analyzed data. D.M., P.K. and D.W. generated organoids. S.R., N.H., T.M. and S.P. gave input on study design. D.D. and J.K. generated the iPrime cell line. S.R., S.P. and T.M. wrote the paper with input from all authors.

## Funding

## Competing interests

Related patent applications on repair gene modified cell lines (patent applicant: Max Planck Society; inventors: S.R. and T.M.; application number: EP17203591.7 and PCT/EP2018/059173; publication: 2018-10-17 WO2018189186; status: pending) and compounds for transient HDR increase (patent applicant: Max Planck Society; inventors: S.R. and T.M.; application number: EP18215071.4; publication: 2020-06-25 WO2020127738A1; status: pending) have been filed. All other authors declare no competing interests.

## Additional information

**Extended data** is available for this paper at https://doi.org/10.1038/s41592-023-01949-1.

**Correspondence and requests for materials** should be addressed to Stephan Riesenberg.

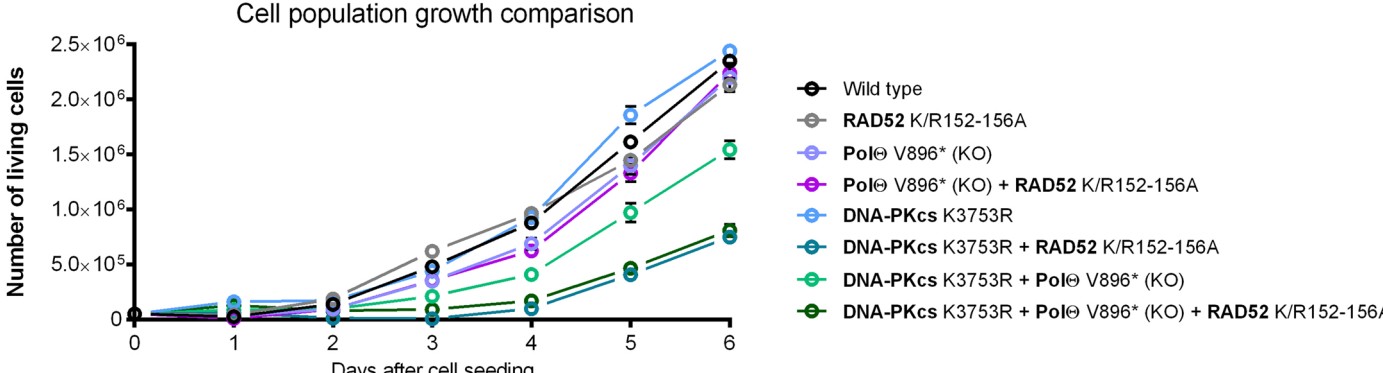

**Extended Data Fig. 1 | Cell population growth.** Number of living wild type H9 hESCs and repair mutant variants for 6 days of cell culture. 50,000 cells each were seeded on day 0. The lines correspond to wild type (black) or repair gene mutants (colored). The amount of living cells was quantified by a fluorescence resazurin assay. Absolute cell number was estimated by comparing resazurin assay fluorescence and cell counting using the Countess Automated Cell Counter (Invitrogen) on day 1. Error bars show the s.e.m. of replicates (n = 2).

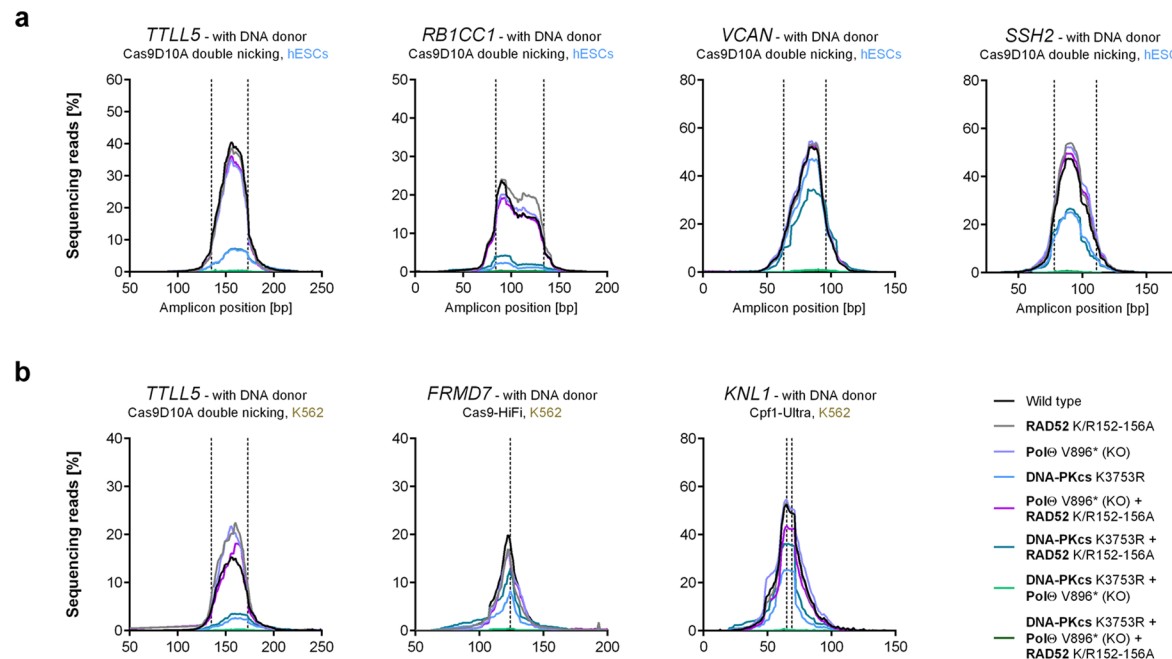

**Extended Data Fig. 2 | Deletion patterns after editing with single-stranded DNA donors.** (**a**) Deletion pattern shapes after editing in hESCs with Cas9D10A double nicking (corresponding to Fig. 1b). (**b**) Deletion pattern shapes after editing in K562 cells with Cas9D10A double nicking, Cas9-HiFi, and Cas12a (Cpf1-Ultra) (corresponding to Fig. 1c). The lines correspond to wild type (black) or repair gene mutants (colored). Each line is the mean of independent biological replicates (n = 3). The vertical dotted line indicates the position of the nicking or cleavage sites.

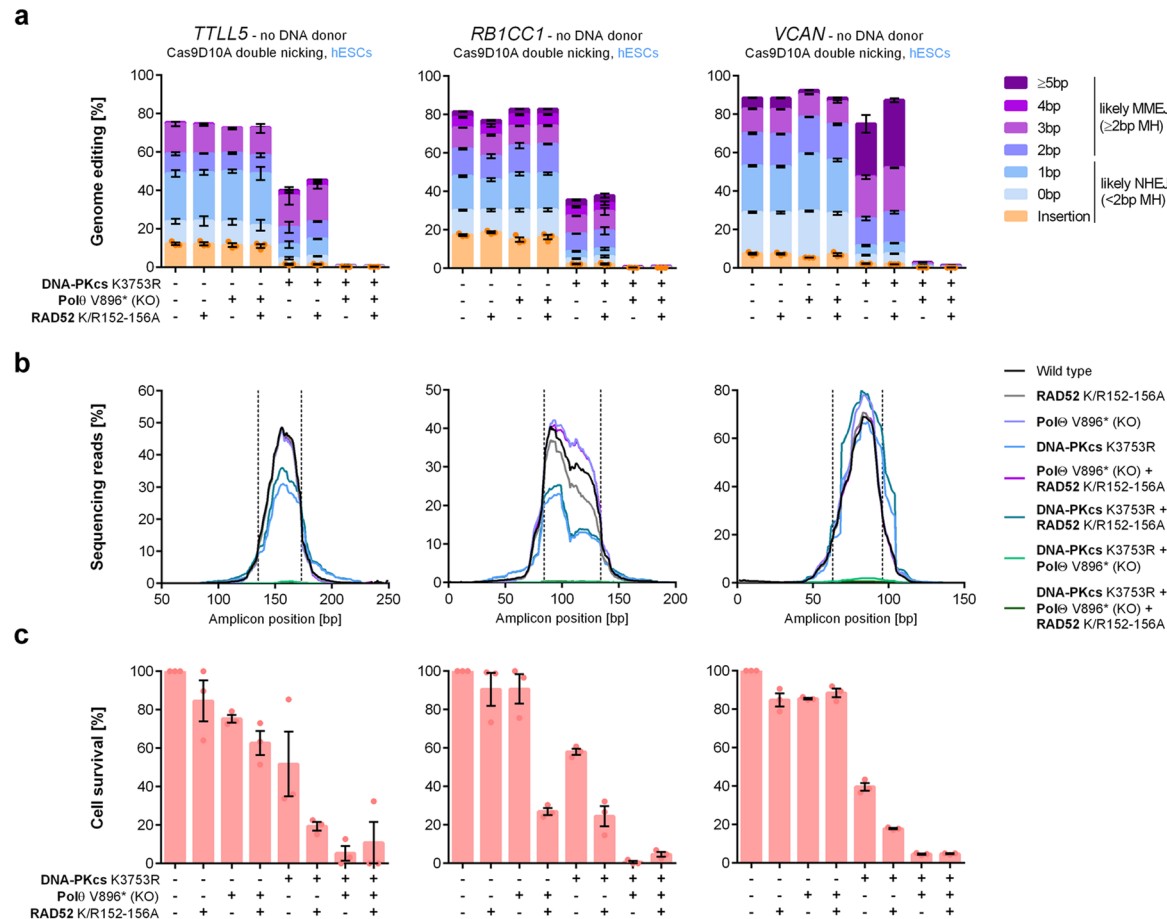

**Extended Data Fig. 3 | Indel efficiencies and cell survival after editing without DNA donors. (a)** Genome editing efficiencies using Cas9D10A double nicking without a DNA donor in H9 hESCs that carry either no repair gene mutation or combinations of DNA-PKcs K3753R, PolΘ V896*, and RAD52 K152A/R153A/R156A (K/R152-156A). Frequencies of deletions are presented based on microhomology (MH) length. For insertions, replicates are depicted by dots. **(b)** Deletion pattern shapes of editing from panel a. The lines correspond to wild type (black) or repair gene mutants (colored). Each line is the mean of replicates. **(c)** Cell survival corresponding to edits from panel **a**. Cell survival was quantified by a fluorescence resazurin assay with respect to editing in wild type cells. Independent biological replicates were performed (n = 3) and error bars show the s.e.m.

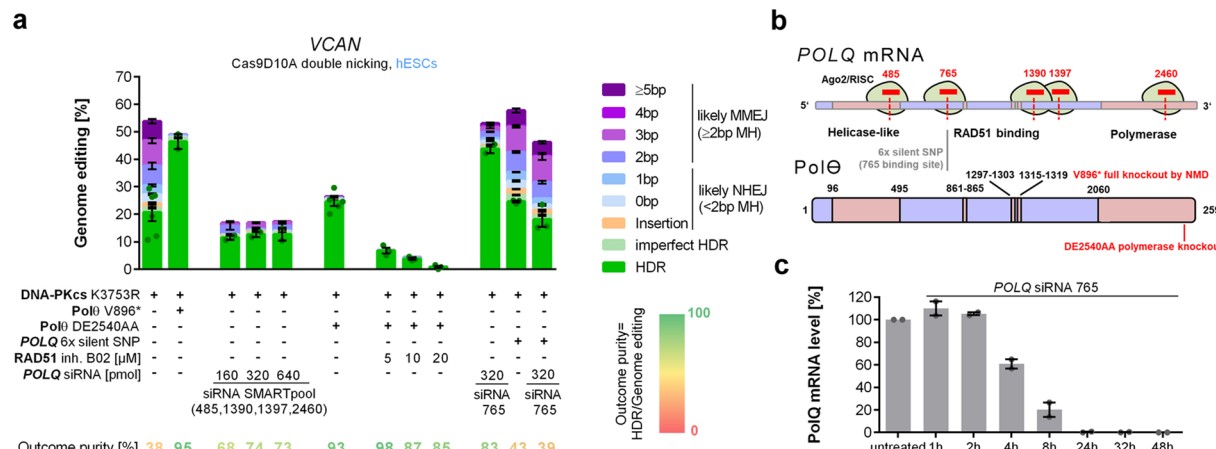

**Extended Data Fig. 4 | Transient microhomology-mediated end-joining (MMEJ) inhibition by siRNAs.** (**a**) Editing efficiencies of the *VCAN* target using Cas9D10A double nicking in H9 hESCs carrying the K3753R mutation, and with additional POLQ mutations (V896* full knockout, DE2540AA polymerase knockout, 6x silent SNPs that do not change the encoding amino acids), transient POLQ inhibition with siRNAs, or transient RAD51 inhibition with the small molecule B02. The K3753R mutation will prevent backup NHEJ repair when MMEJ is inhibited. Frequencies of deletions are presented based on microhomology (MH) length. For HDR, replicates are depicted by dots. The mean outcome purities (percentage HDR of all editing events) are given. Independent biological replicates were performed (n = 2, except DNA-PKcs K3753R / DE2540AA / B02

and DNA-PKcs K3753R / 6x silent SNP n = 3, DNA-PKcs K3753R / DE2540AA n = 5, and DNA-PKcs K3753R n = 6) and error bars show the s.e.m. (**b**) A scheme of the different siRNAs targeting the *POLQ* mRNA to induce Ago2/RISC assisted cleavage as well as a site of silent mutations that do not change the encoded amino acids, and translated PolΘ is shown below. PolΘ motifs described to be detrimental for homologous recombination/HDR are colored rose, PolΘ inhibitory mutations are red. (**c**) Time course of *POLQ* mRNA knock down with siRNA 765. *POLQ* expression was normalized with *GAPDH* expression and is given relative to untreated cells. Independent biological replicates were performed (n = 2) and error bars show the s.e.m.

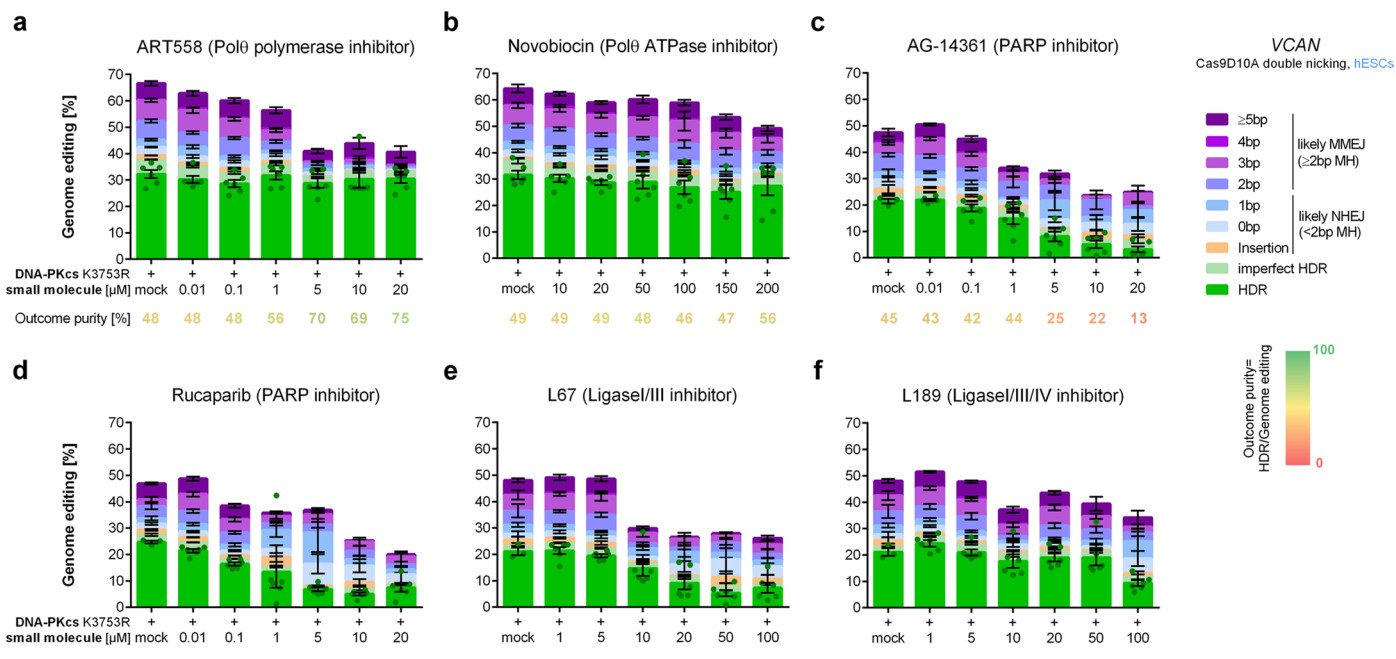

**Extended Data Fig. 5 | Transient microhomology-mediated end-joining (MMEJ) inhibition by small molecules.** Editing efficiencies of the *VCAN* target using Cas9D10A double nicking in H9 hESCs carrying the K3753R mutation and additional small molecules to inhibit the MMEJ repair proteins PolΘ, PARP or Ligase I/III: (**a**) ART558, (**b**) Novobiocin, (**c**) AG-14361, (**d**) Rucaparib, (**e**) L67,

(**f**) L189. Frequencies of deletions are presented based on microhomology (MH) length. For HDR, replicates are depicted by dots. The mean outcome purities (percentage HDR of all editing events) are given. Independent biological replicates were performed (n = 6, except n = 3 for mock c-f) and error bars show the s.e.m.

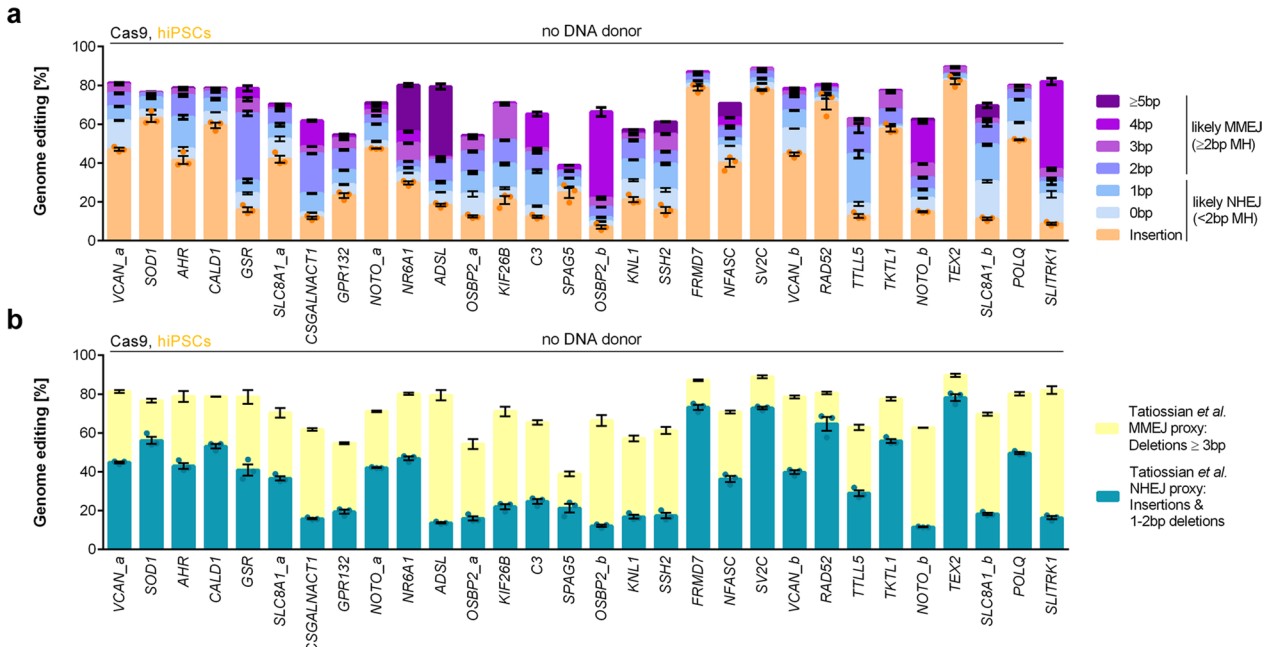

**Extended Data Fig. 6 | Indel signatures of no donor control edits with Cas9.**
Indel signatures after genome editing of thirty targets using iCRISPR-Cas9 in
409B2 hiPSCs without a DNA donor. Indels are plotted in two ways: (**a**) Insertions
are indicated in orange, and deletions with various microhomology (MH) lengths
are indicated with various shades of blue and purple. For insertions, replicates
are depicted by dots. (**b**) Deletions with sizes equal or bigger 3 bp are indicated
in yellow, while all other indels are indicated in petrol with replicates as dots. The
former and latter have been described as proxies for NHEJ and MMEJ after Cas9
editing, respectively, and a strong proxy for MMEJ is predictive of tendency for
HDR if a DNA donor would be present[40]. These no donor controls are related to
Fig. 2d. Independent biological replicates were performed (n = 3) and error bars
show the s.e.m.

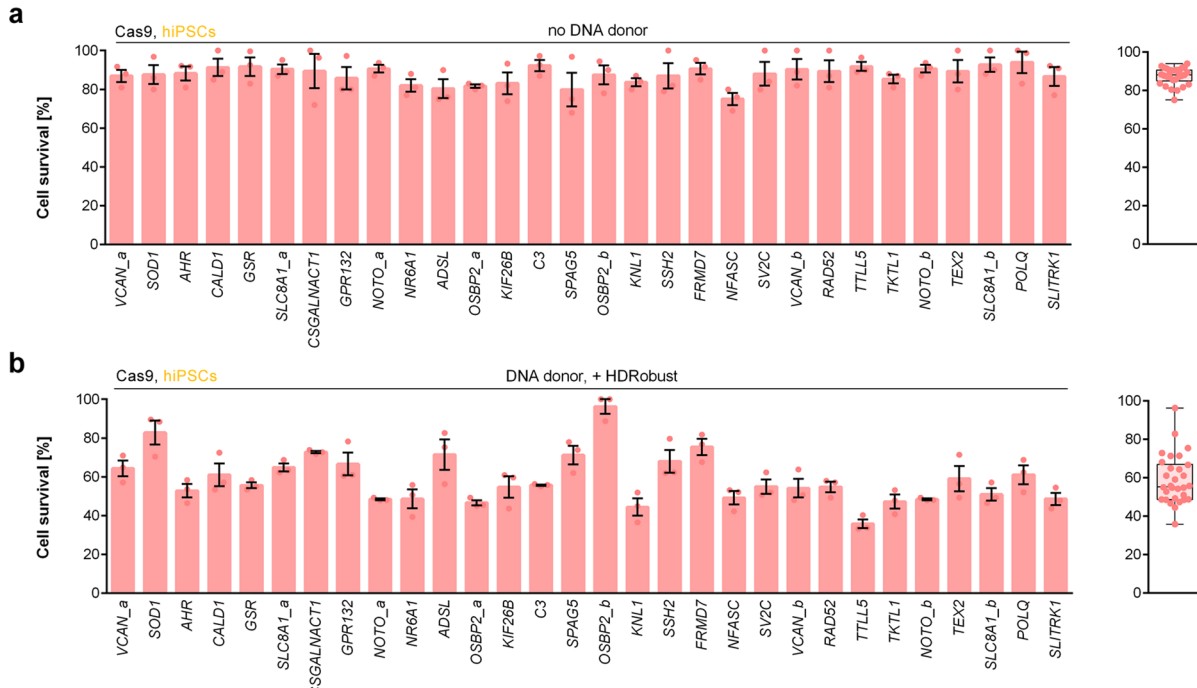

**Extended Data Fig. 7 | Cell survival after Cas9 editing for indel generation without a DNA donor and precise editing with HDRobust.** Cell survival after Cas9 edits of thirty targets in 409B2 hiPSCs without DNA donors (related sequencing data Extended Data Fig. 6) (**a**), and with both DNA donors and HDRobust (related sequencing data Fig. 2d) (**b**). Cell survival was quantified by a fluorescence resazurin assay with respect to mock electroporation without editing. Independent biological replicates are depicted by dots (n = 3) and error bars show the s.e.m. Each dot in the right panels indicate the mean of one target, the box the 25th to 75th percentile, the line the median and whiskers extend to the minimum and maximum values.

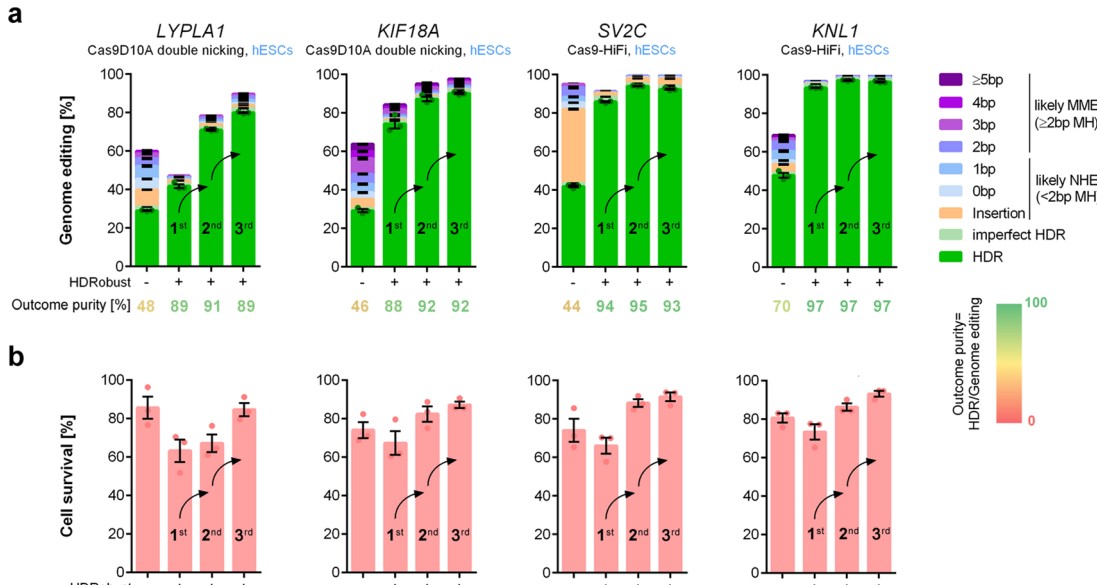

**Extended Data Fig. 8 | Repeated editing using HDRobust to enrich for HDR edited cells.** (**a**) Genome editing efficiencies using repeated Cas9D10A double nicking or Cas9-HiFi in combination with HDRobust in H9 hESCs that carry no repair gene mutation. Frequencies of deletions are presented based on microhomology (MH) length. For HDR, replicates are depicted by dots. The mean outcome purity given below is the percentage of HDR of all editing events. (**b**) Cell survival corresponding to edits from panel **a**. Cell survival was quantified by a fluorescence resazurin assay with respect to editing in wild type cells. Independent biological replicates were performed (n = 3) and error bars show the s.e.m.

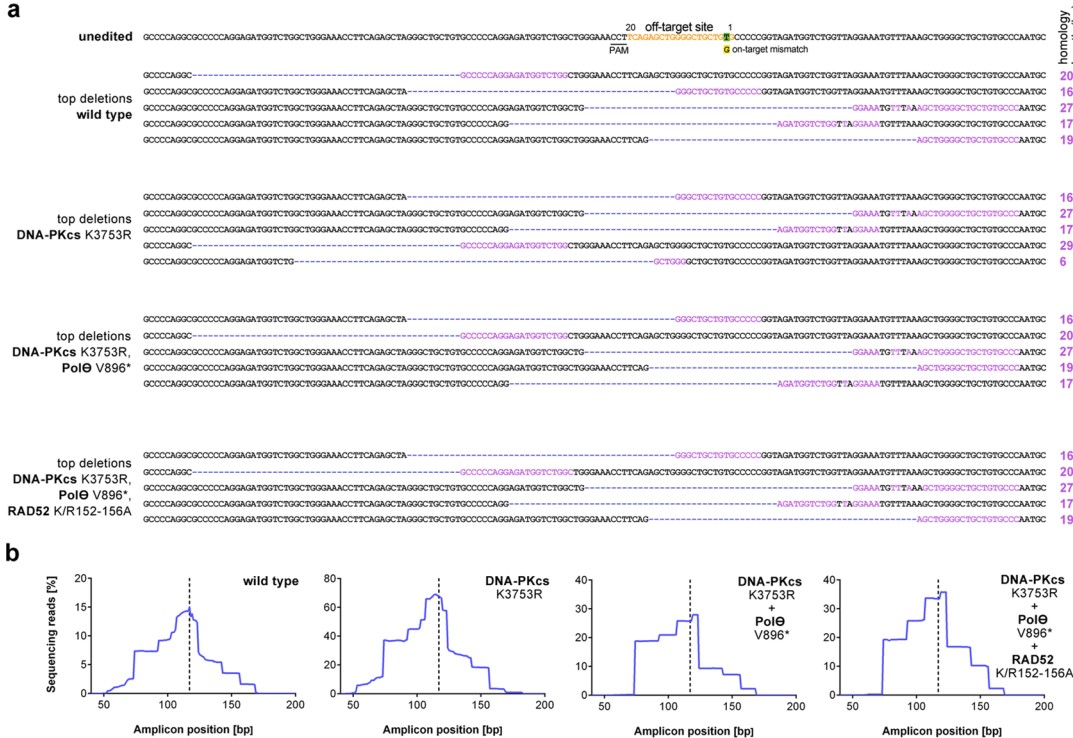

**Extended Data Fig. 9 | Deletion events at the OSBP2 off-target *O*-OT-1 in cell lines with repair gene mutant combinations.** (**a**) Top five deletions at the *OSBP2* off-target *O*-OT-1 H9 hESCs that carry no repair gene mutation or combinations of DNA-PKcs K3753R, PolΘ V896*, and RAD52 K152A/R153A/R156A (K/R152-156A). Sequence similarity flanking deletions indicated in purple and the numbers of similar bases ('homology length') are given. (**b**) Deletion frequency

at the *OSBP2* off-target *O*-OT-1 H9 hESCs that carry no repair gene mutation or combinations of DNA-PKcs K3753R, PolΘ V896*, and RAD52 K/R152-156A. The purple deletion line is the mean of independent biological replicates (n = 3). The vertical dotted line indicates the position of the off-target cut. Data shown is related to Fig. 4c.

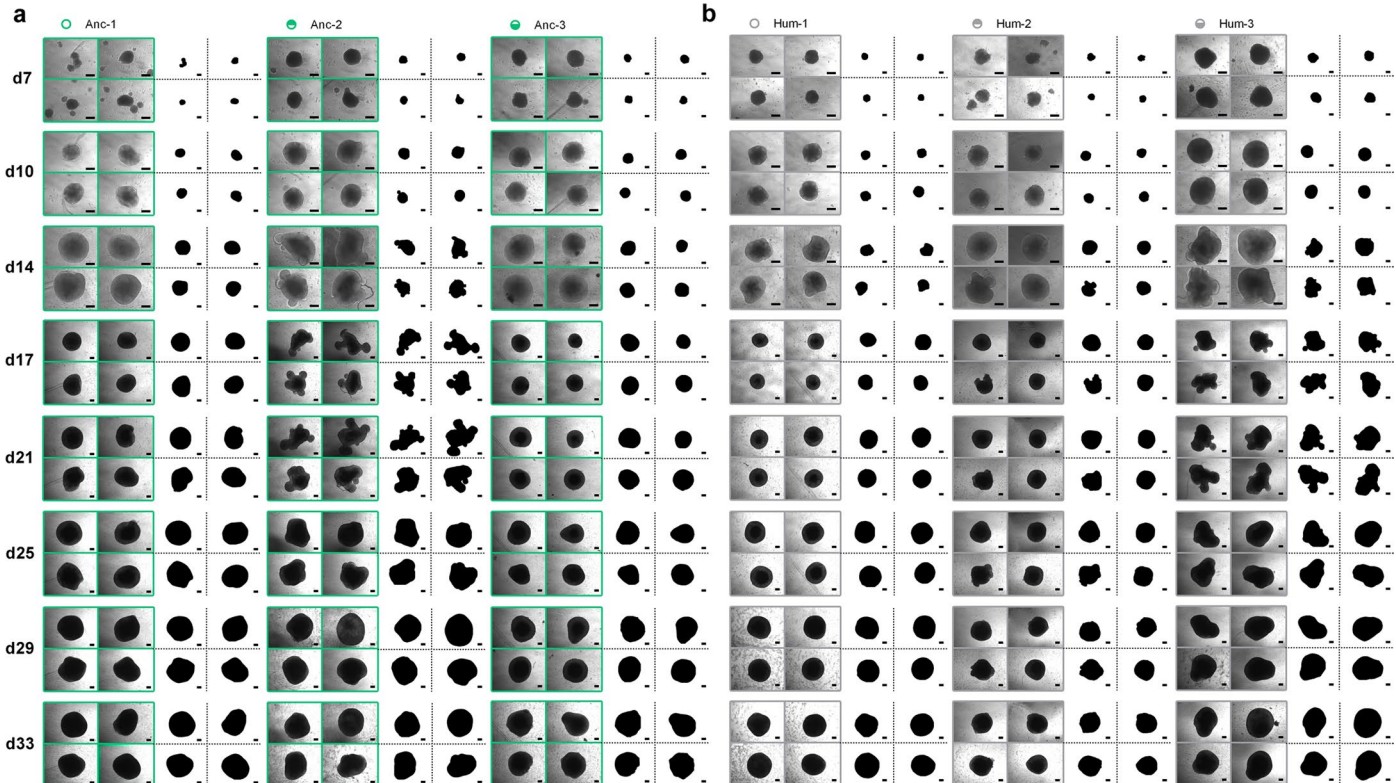

**Extended Data Fig. 10 | *NOVA1* brain organoid images used for area and shape analysis.** Phase-contrast images and shapes of brain organoids derived from cellular clones carrying ancestral (**a**) or human (**b**) *NOVA1*. The time course of organoid development from d7 to d33 (top to bottom) of three different cellular clones for and ancestral (Anc1-3, green circles and image frames) and human (Hum1-3, gray circles and image frames) *NOVA1* is shown for four different organoids for each day and clone. The size bar is 200 µm.

# Reporting Summary

## Statistics

For all statistical analyses, confirm that the following items are present in the figure legend, table legend, main text, or Methods section.

| n/a | Confirmed | |
|---|---|---|
| ☐ | ☒ | The exact sample size (*n*) for each experimental group/condition, given as a discrete number and unit of measurement |
| ☐ | ☒ | A statement on whether measurements were taken from distinct samples or whether the same sample was measured repeatedly |
| ☒ | ☐ | The statistical test(s) used AND whether they are one- or two-sided *Only common tests should be described solely by name; describe more complex techniques in the Methods section.* |
| ☒ | ☐ | A description of all covariates tested |
| ☒ | ☐ | A description of any assumptions or corrections, such as tests of normality and adjustment for multiple comparisons |
| ☐ | ☒ | A full description of the statistical parameters including central tendency (e.g. means) or other basic estimates (e.g. regression coefficient) AND variation (e.g. standard deviation) or associated estimates of uncertainty (e.g. confidence intervals) |
| ☒ | ☐ | For null hypothesis testing, the test statistic (e.g. *F*, *t*, *r*) with confidence intervals, effect sizes, degrees of freedom and *P* value noted *Give P values as exact values whenever suitable.* |
| ☒ | ☐ | For Bayesian analysis, information on the choice of priors and Markov chain Monte Carlo settings |
| ☒ | ☐ | For hierarchical and complex designs, identification of the appropriate level for tests and full reporting of outcomes |
| ☒ | ☐ | Estimates of effect sizes (e.g. Cohen's *d*, Pearson's *r*), indicating how they were calculated |

*Our web collection on statistics for biologists contains articles on many of the points above.*

## Software and code

Policy information about availability of computer code

| Data collection | No software was used. |
|---|---|
| Data analysis | CRISPResso (v1), SAMtools (v1.12), GraphPad Prism (v6), Adobe Photoshop (CS5), ImageJ (v1.53) |

For manuscripts utilizing custom algorithms or software that are central to the research but not yet described in published literature, software must be made available to editors and reviewers. We strongly encourage code deposition in a community repository (e.g. GitHub). See the Nature Portfolio guidelines for submitting code & software for further information.

## Data

Policy information about availability of data

All manuscripts must include a data availability statement. This statement should provide the following information, where applicable:

- Accession codes, unique identifiers, or web links for publicly available datasets
- A description of any restrictions on data availability
- For clinical datasets or third party data, please ensure that the statement adheres to our policy

The sequencing data generated in this study have been deposited in the Dryad database under accession code dryad.fj6q5740f. Data are also available on request from authors.

## Human research participants

Policy information about studies involving human research participants and Sex and Gender in Research.

| | |
|---|---|
| Reporting on sex and gender | N/A |
| Population characteristics | N/A |
| Recruitment | N/A |
| Ethics oversight | N/A |

Note that full information on the approval of the study protocol must also be provided in the manuscript.

# Field-specific reporting

Please select the one below that is the best fit for your research. If you are not sure, read the appropriate sections before making your selection.

☒ Life sciences  ☐ Behavioural & social sciences  ☐ Ecological, evolutionary & environmental sciences

For a reference copy of the document with all sections, see nature.com/documents/nr-reporting-summary-flat.pdf

# Life sciences study design

All studies must disclose on these points even when the disclosure is negative.

| | |
|---|---|
| Sample size | No sample size calculation was performed. At least two independent biological replicates for each condition were obtained. This was sufficient as the editing efficiency difference between replicates of a respective group was very small compared to the strong difference observed between the compared groups. |
| Data exclusions | No data were exluded from analysis. |
| Replication | All attempts of replication were successful. At least two independent biological replicates for each condition were obtained. |
| Randomization | Mammalian cells used in this study were cells grown under identical conditions and experiments were done in parallel. Thus, no randomization was used. |
| Blinding | Mammalian cells used in this study were cells grown under identical conditions and experiments were done in parallel. Thus, blinding was not used for data collection. Analysis was performed based on numerical sample names, without the identity of the samples known during the analysis. |

# Reporting for specific materials, systems and methods

We require information from authors about some types of materials, experimental systems and methods used in many studies. Here, indicate whether each material, system or method listed is relevant to your study. If you are not sure if a list item applies to your research, read the appropriate section before selecting a response.

### Materials & experimental systems

| n/a | Involved in the study |
|---|---|
| ☒ ☐ | Antibodies |
| ☐ ☒ | Eukaryotic cell lines |
| ☒ ☐ | Palaeontology and archaeology |
| ☒ ☐ | Animals and other organisms |
| ☒ ☐ | Clinical data |
| ☒ ☐ | Dual use research of concern |

### Methods

| n/a | Involved in the study |
|---|---|
| ☒ ☐ | ChIP-seq |
| ☒ ☐ | Flow cytometry |
| ☒ ☐ | MRI-based neuroimaging |

## Eukaryotic cell lines

Policy information about cell lines and Sex and Gender in Research

| | |
|---|---|
| Cell line source(s) | H9 hESCs (WiCell Research Institute, WA09), 409-B2 hiPSCs (Riken BioResource Center, HPS0076), K562 cells (ECACC, |

| Cell line source(s) | 89121407), CD4+ T cells (HemaCare, PB04C-1), Lymphoblastoid cells (Coriell Institute, GM14890, HG02367, GM16265, GM08369) |
| --- | --- |
| Authentication | All cell lines were authenticated by the supplier via certificate of analysis and additionally in-house by checking morphology. |
| Mycoplasma contamination | All cell lines were tested negative for mycoplasma contamination before and after the experiments. |
| Commonly misidentified lines (See ICLAC register) | No commonly misidentified cell lines were used in the study. |

