## [Peer Review File · Nature Methods]

Peer Review Information

Manuscript Title: Efficient High Precision HDR-dependent Genome Editing by HDRobust

Corresponding author name(s): Stephan Riesenberg

Editorial Notes: n/a

Reviewer Comments & Decisions:

Decision Letter, initial version:

Dear Dr Riesenberg,

Your Article entitled "Predominant HDR-dependent genome editing and prevention of byproducts by DNA repair pathway choice" has now been seen by 3 reviewers, whose comments are attached. While they find your work of potential interest, they have raised serious concerns which in our view are sufficiently important that they preclude publication of the work in Nature Methods, at least in its present form.

As you will see, the reviewers raise serious concerns about the mechanistic interpretations presented and the potential genotoxicity that may be introduced during inhibitions.

Should further experimental data allow you to fully address these criticisms we would be willing to look at a revised manuscript (unless, of course, something similar has by then been accepted at Nature Methods or appeared elsewhere). This includes submission or publication of a portion of this work somewhere else. We hope you understand that until we have read the revised paper in its entirety we cannot promise that it will be sent back for peer-review.

If you are interested in revising this manuscript for submission to Nature Methods in the future, please contact me to discuss your appeal before making any revisions. Otherwise, we hope that you find the reviewers' comments helpful when preparing your paper for submission elsewhere.

Best regards,
Lei

Lei Tang, Ph.D.
Senior Editor
Nature Methods

Reviewers' Comments:

Reviewer #1:

Remarks to the Author:

Riesenberg et al in "Predominant HDR-dependent genome editing and prevention of byproducts by DNA repair pathway choice" describe their work on inhibiting two pathways of end-joining to enhance HDR editing. This work builds on prior work in which they inhibited canonical NHEJ with the small molecule M3814. In this work they inhibit DNA-PKcs and PolQ. They also genetically inactivate RAD52 as part of the work and interestingly that seemed to have little affect (not even on viability). They use the ssODN method for HDR in a variety of cell types including human ESC, iPSC, K562 and some T-cell work. Overall, they make the interesting observation that the inhibition of both pathways can increase the "purity" of the edits--a term they use to describe the fraction of edits that are HDR rather than NHEJ or MMEJ mediated edits. They describe their transient inhibition of the two pathways "HDRrobust" using a combination of small molecules and siRNA. I appreciate the authors comparison to prime Editing and the conclusions drawn from the data about the two systems. Finally, they perform some assays for genotoxicity and surprisingly find that the genotoxicity seems to be reduced. This reduction is somewhat reassuring but also raises concerns that the assays used are missing something Overall, the work is carefully done and represents an important advance. The following, however, are important considerations for the authors to address:

1. As seems to be the norm, the authors ignore the work from multiple labs including the Porteus, Cannon, Naldini, Malech labs showing that high frequencies of HDR can be achieved in primary human cells, including IPS and ES cells, using HiFi RNP Cas9/gRNA and AAV6. In the AAV6 donor system more than single nucleotide changes can be made as well. What is strange is the authors do cite the Vakulskas et al paper where high frequencies of HDR were reported across multiple loci. The authors need to address this common oversight. In the discussion, they need to mention that they have not tested their system using the most robust and most versatile system of HDR using RNP and AAV6 as the donor.
2. The genetic mutations of DNA-PKcs and PolQ are interesting POC but it is the transient inhibition that has the potential to make the most impact.
3. Did they look at the degree and timing of when PolQ is knocked down with the siRNA. It seems like the timing might not match the editing
- 4 There are small molecules reported to inhibit PolQ, did the authors study those? siRNA knockdown can be non-specific and problematic in many circumstances--especially in RNP based Cas9 editing.

5 The inhibition of PolQ alone did not seem to result in a reduction of MMEJ defined INDELS. This suggests that the mechanism by which it is acting is more complicated than the authors suggest.

6. It would be helpful for a more detailed communication of the frequency and INDEL type at different targets rather than lumping into "NHEJ" vs "MMEJ" INDELS. The effect of inhibition of the different pathways on each type of INDEL should be more fully analyzed.

7. As noted above, the lack of genotoxicity is surprising as both pathways are needed to maintain genomic integrity in cells. It is possible that the dual inhibition is creating cellular toxicity through unrepaired DNA breaks and creating significant genomic rearrangements leading to cell death. Thereby potentially "purifying" the population. The point being that on-target, off-target, and spontaneous breaks all have to be repaired somehow and how are they being repaired with dual pathway inhibition? The authors should show both short and long-term cell viability data. It is possible that high HDR purity is being achieved at the cost of low cell viability. In sum, the authors have done some good analysis of the genotoxic effects of the dual inhibition but there is more to be done given the relationship between intact DNA repair pathways maintaining genomic integrity and the inhibition they do here.

Reviewer #2:

Remarks to the Author:

In their manuscript, "Predominant HDR-dependent genome editing and prevention of byproducts by DNA repair pathway choice," Riesenbergs and colleagues investigate the efficiency and purity of programmed genome editing after inhibiting various DNA double-strand break repair pathways. Their results establish that simultaneous interference of non-homologous end-joining and alternative end-joining (by mutating, inhibiting, and/or depleting the key DNA repair factors DNA-PKcs and POLQ) dramatically improves the purity and, in some cases, the efficiency of installing single nucleotide variants with programmable nucleases and single-strand oligonucleotide DNA donors. This work builds off of previous results using only interference of DNA-PKcs to improve genome editing by the same authors (PMID: 31392986) and others (PMID: 26307031, 21300785). The specific advance described here, that combination of DNA-PK inhibition with loss of POLQ reduces indel byproducts during genome editing, is a moderate but notable finding, and the authors have done a nice job defining the scope of the observation (i.e., evaluated many target sites and modalities of editing). Given the site-to-site variability of overall HDR efficiency observed, however, the authors fall short of demonstrating broad utility of the approach as a method. Moreover, they provide an oversimplified mechanistic interpretation of their results. Nevertheless, with some additional controls, important corrections, and improved exposition, this would be a useful study.

Major comments:

- As a methods paper, the major shortcoming of this manuscript is that, while the authors show that combined interference of DNA-PKcs and POLQ activity increases the purity of installing single-nucleotide variants into the genome, how well that finding will translate into practical utility remains unclear: (1) The most useful version of their approach is what they call HDRobust (transient inhibition). How well cells will survive such treatment, however, is an open question. Indeed, loss of POLQ has been shown to be synthetic lethal with many DNA repair genes—including NHEJ factors (PMID: 31537809)—and the authors demonstrate that their mutant cell lines have strong cell growth phenotypes during editing (Supplementary Fig. 3). The same cell survival analysis should be performed and reported for HDRobust. (2) siRNAs are well known to have profound off-target effects (PMID: 20043028) and using five siRNAs to transiently inhibit POLQ is likely to be problematic for some applications (and is technically unnecessary). If RNAi-mediated depletion of POLQ is indeed the most robust strategy for this approach, a single siRNA that can perform well should be identified and described (and rescue of that effect by reintroduction of an siRNA-resistant POLQ should be demonstrated). Alternatively, small molecule inhibitors of POLQ's polymerase activity have been recently described (PMID: 34140467). Testing of these would be a valuable addition to this paper. (3) One of the more exciting results in the paper is the finding that repeated editing with the authors' approach can be used to edit a bulk cell population to near uniformity "without the isolation of cellular clones" (Figure 2e). However, despite a broad claim that HDRobust is well suited for this purpose, this particular feature is demonstrated at only one edit site (and without analysis of cell viability during editing). I'd recommend providing data to address (1) and (2), as well as expanding the examples of (3).

- A second major concern is that the paper oversimplifies our current mechanistic understanding of DSB repair at nuclease induced double-strand breaks:

Example 1: The authors describe HDR as a single mechanistic process without discriminating between the forms of HDR they deploy in the paper (i.e., single-strand template repair, SSTR, and polymerase theta-mediated end-joining, TMEJ) and HDR mechanisms not evaluated (e.g., homologous recombination, HR). Ultimately, I'd argue that such oversimplifications do not necessarily diminish the actual results in the manuscript, but they do cause interpretation in the paper to be muddled and, at times, incorrect. Most notably, it's important to point out that SSTR has been shown to be RAD51-independent in many studies, albeit not all (PMID: 16908537, 28067217, 33057349, 34672952). Given this observation, it is a glaring oversight for the authors to discuss RAD51 as definitive mediator of SSTR (aka "HDR") throughout. Of course, with this in mind, speculation "that the siRNA-induced cleavage of the mRNA" results "in a truncated proteins that may sequester RAD51 and thereby inhibit HDR" seems very unlikely in their experiments. I'd strongly recommend that the authors provide a more scholarly description of SSTR and either rescue the effect of their siRNA pool by reintroduction of an siRNA-resistant POLQ or provide a clear discussion outlining the caveats associated with inferring mechanism from the provided results.

Example 2: At least in yeast, SSTR is Rad52-dependent (PMID: 33057349). How the RAD52 mutations examined in the paper would be expected to impact SSTR is therefore not at all clear a priori. I'd strongly recommend that the authors provide a more nuanced description of published data regarding the potential role of Rad52 in SSTR and other DSB repair pathways.

- A third major concern centers on the authors' comparison of HDRobust to prime editing. The authors claim in their discussion that "editing using a DNA donor and HDRobust outperforms even optimized PE in terms of absolute precise editing efficiency (up to 91% HDR) as well as outcome purity (up to 96%)." This comparison is problematic for two reasons: First, an enormous amount of work has recently gone into optimizing prime editing systems (e.g., PMID: 34608327), but, rather than use the most optimized editors, the authors appear to have generated their own prime editing systems to facilitate comparisons. Moreover, some of the published prime editing rates provided in Figure 4b are clearly not from the most efficient nor the most precise method(s) of prime editing available (indeed, some even appear to have been selected from figures where better rates are included in the same figure, PMID: 34608327). In order to make a direct and fair comparison, authors must use state-of-the-art approaches/data.

Minor comments:

- The abstract of this paper claims that "up to 91% of chromosomes in populations of cells will carry desired point mutations introduced by HDR" and the authors reiterate this "up to 91%" claim again in the discussion. In my view, such "up to" claims serve far less good than just reporting the actual range of editing observed.
- The diagrams indicating editing conditions (nucleases and guides) throughout the manuscript are confusing. Clarity would be improved by just using words to indicate conditions instead of cartoons.
- Recent work has shown that the presence of a single-strand oligo donor can impact TMEJ (PMID: 33160457, 34672952). Inclusion of no donor controls would therefore be a valuable addition to this paper.
- For some reason, the authors relegate some key results to a Supplementary Discussion section rather than included them in the main text (e.g., effects on cell viability). I would recommend these data be included in the main text and figures.

Reviewer #3:

Remarks to the Author:

The submitted manuscript presents an impressive body of work aimed at a long-standing goal in the field of genome editing: maximizing a particular outcome from the DSB (or the nick, or a prime-edit) – the introduction of a repair-template-specified mutation. A great wealth of studies have been devoted to this issue over the past 17 years, and it's safe to say that the kinds of numbers the authors are reporting in terms of HDR percentage are some of the best ever seen, and the work will be of interest to the large scientific community of genome editors.

Comments / questions

The authors use a “nickase” form of Cas9 and two gRNAs to drive editing in “dual-nickase” mode. Do the authors have any data on single-nickase editing? As the authors are well-aware, there is a 10 year-long precedent (work from Sangamo and Keith Joung, among others; pubmed IDs 22434427 and 22373919) that a such a single-nickase approach markedly reduces non-HDR-driven outcomes.

It is unclear from the Materials and Methods whether any optimization of donor concentration was attempted. As the authors know, starting with the first report of using ssODNs for genome editing via HDR (2011 paper from Sigma-Aldrich, pubmed ID 21765410), it's been clear that titrating ODN quantity can improve the HDR/NHEJ ratio.

It is unclear whether the authors introduced blocking mutations into the donor to prevent recleavage of the HDR-repaired template – another established way to limit NHEJ/MMEJ-based outcomes.

There is a striking disparity in Fig 1b between the first two genes and the second two. In the former, all the edited alleles convert to HDR ones (in other words, the absolute editing efficiency stays the same, and all the editing is resolved by HDR). For the second two, that is very clearly not the case. The absolute editing efficiency drops. Do the authors have any insight as to why?

Do the authors have any data on whether HDRobust affects hESC or iPSC stemness / totipotency? In other words, does the pharmacological inhibition of NHEJ and siRNA inhibition of SSA create an epigenetic memory that limits the ability to differentiate the stem cells into various progeny?

The cell survival discussion should be moved from supplementary to primary – it's an important issue. In particular the authors should draw the reader's attention to the fact that “dirty” guides will not work with HDRobust and instead the edited cells will just die.

Author Rebuttal to Initial comments

Point-by-point revision plan

Reviewers' Comments:

Reviewer #1:

Remarks to the Author:

Riesenberg et al in "Predominant HDR-dependent genome editing and prevention of byproducts by DNA repair pathway choice" describe their work on inhibiting two pathways of end-joining to enhance HDR editing. This work builds on prior work in which they inhibited canonical NHEJ with the small molecule M3814. In this work they inhibit DNA-PKcs and PolQ. They also genetically inactivate RAD52 as part of the work and interestingly that seemed to have little affect (not even on viability). They use the ssODN method for HDR in a variety of cell types including human ESC, iPSC, K562 and some T-cell work. Overall, they make the interesting observation that the inhibition of both pathways can increase the "purity" of the edits--a term they use to describe the fraction of edits that are HDR rather than NHEJ or MMEJ mediated edits. They describe their transient inhibition of the two pathways "HDRrobust" using a combination of small molecules and siRNA. I appreciate the authors comparison to prime Editing and the conclusions drawn from the data about the two systems. Finally, they perform some assays for genotoxicity and surprisingly find that the genotoxicity seems to be reduced. This reduction is somewhat reassuring but also raises concerns that the assays used are missing something. Overall, the work is carefully done and represents an important advance.

We thank the reviewer for the kind and encouraging words. With regard to genotoxicity, see our comments on point 7 below.

The following, however, are important considerations for the authors to address:

1. As seems to be the norm, the authors ignore the work from multiple labs including the Porteus, Cannon, Naldini, Malech labs showing that high frequencies of HDR can be achieved in primary human cells, including IPS and ES cells, using HiFi RNP Cas9/gRNA and AAV6. In the AAV6 donor system more than single nucleotide changes can be made as well. What is strange is the authors do cite the Vakulskas et al paper where high frequencies of HDR were reported across multiple loci. The authors need to address this common oversight. In the discussion, they need to mention that they have not tested their system using the most robust and most versatile system of HDR using RNP and AAV6 as the donor.

We apologize for this oversight. We will mention that our approach likely also helps to further increase the already high efficiencies achieved when using RNP and AAV6 donor, even though we did not test it experimentally. We will cite the related literature.

2. The genetic mutations of DNA-PKcs and PolQ are interesting POC but it is the transient inhibition that has the potential to make the most impact.

We agree. The genetic mutations primarily serve the purpose to exclude that any unknown effects of the reagents used in the transient inhibition would cause the effects observed.

3. Did they look at the degree and timing of when PolQ is knocked down with the siRNA. It seems like the timing might not match the editing

We will add a time course of POLQ knockdown by siRNA.

4 There are small molecules reported to inhibit PolQ, did the authors study those? siRNA knockdown can be non-specific and problematic in many circumstances--especially in RNP based Cas9 editing.

We will test small molecule inhibitors of POLQ (ART558 and Novobiocin) as also suggested by reviewer 2. In addition, the genetic knockout of POLQ demonstrates that it increases the editing efficiency in combination with NHEJ inhibition by DNA-PKcs K3753R.

5 The inhibition of PolQ alone did not seem to result in a reduction of MMEJ defined INDELS. This suggests that the mechanism by which it is acting is more complicated than the authors suggest.

This is likely due to the ad hoc approach we use to score indels as caused by NHEJ and by MMEJ. When deletions occurred at sites where at least two nucleotides on one end of the deletion was identical to two nucleotides in the undeleted sequence on the other end we scored this as MMEJ. However, some of these deletions could also be due to NHEJ that occurred at these positions by chance. We will do an additional Indel pattern analysis (see comment below) and will also investigate if reanalyzing the data defining MMEJ as deletions of at least 3 nucleotides is a better predictor of MMEJ vs. NHEJ (as suggested in Tatiassian et al., Mol. Ther. 29:1057-69 3:29 (2021) PMID 33160457).

6. It would be helpful for a more detailed communication of the frequency and INDEL type at different targets rather than lumping into "NHEJ" vs "MMEJ" INDELS. The effect of inhibition of the different pathways on each type of INDEL should be more fully analyzed.

We will add additional analysis of Indel types and show Indel size patterns for different pathway inhibitions as done in Supp. Fig. 4b.

7. As noted above, the lack of genotoxicity is surprising as both pathways are needed to maintain genomic integrity in cells. It is possible that the dual inhibition is creating cellular toxicity through unrepaired DNA breaks and creating significant genomic rearrangements leading to cell death. Thereby potentially "purifying" the population. The point being that on-target, off-target, and spontaneous breaks all have to be repaired somehow and how are they being repaired with dual pathway inhibition?

We will extend the discussion by pointing out that cells lacking both NHEJ and MMEJ will (as far as is known) only be able to repair breaks at the on-target sites using the exogenous DNA donor and HDR, and using the sister chromatids and homologous recombination for breaks at off-target sites resulting in perfect repair to the wild type state. If breaks cannot be repaired in this way, cells will die resulting in a population of precisely edited or wild type cells.

In the discussion, we will also point out that inhibition of NHEJ and MMEJ is suitable for cells, but not for editing in organisms, except perhaps in some experimental situations or generation of animal models.

The authors should show both short and long-term cell viability data. It is possible that high HDR purity is being achieved at the cost of low cell viability. In sum, the authors have done some good analysis of the genotoxic effects of the dual inhibition but there is more to be done given the relationship between intact DNA repair pathways maintaining genomic integrity and the inhibition they do here.

We will add data on short-term cell survival when using the HDRobust mix as also suggested by reviewer 2.

However, we believe our data on long-term genotoxic effects (Supp. Fig. 5) show that dual inhibition of both NHEJ and MMEJ does not result in any stark differences of genome stability in the surviving cell population. This is especiall the case since the long-term data in Supp. Fig. 5 shows the results of NHEJ and MMEJ continuous inhibition over 5 months (!) whereas the HDRobust approach inhibits these pathways over only 2 days.

Another line of evidence for limited effect of HDRobust on genomic integrity is our ability to generate cerebral brain organoids after editing (see second last comment to reviewer 3).

Reviewer #2:

Remarks to the Author:

In their manuscript, “Predominant HDR-dependent genome editing and prevention of byproducts by DNA repair pathway choice,” Riesenbergs and colleagues investigate the efficiency and purity of programmed genome editing after inhibiting various DNA double-strand break repair pathways. Their results establish that simultaneous interference of non-homologous end-joining and alternative end-joining (by mutating, inhibiting, and/or depleting the key DNA repair factors DNA-PKcs and POLQ) dramatically improves the purity and, in some cases, the efficiency of installing single nucleotide variants with programmable nucleases and single-strand oligonucleotide DNA donors. This work builds off of previous results using only interference of DNA-PKcs to improve genome editing by the same authors (PMID: 31392986) and others (PMID: 26307031, 21300785). The specific advance described here, that combination of DNA-PK inhibition with loss of POLQ reduces indel byproducts during genome editing, is a moderate but notable finding, and the authors have done a nice job defining the scope of the observation (i.e., evaluated many target sites and modalities of editing). Given the site-to-site variability of overall HDR efficiency observed, however, the authors fall short of demonstrating broad utility of the approach as a method. Moreover, they provide an oversimplified mechanistic interpretation of their results. Nevertheless, with some additional controls, important corrections, and improved exposition, this would be a useful study.

We are glad the reviewer acknowledges our evaluation of many target sites and modalities of editing. In our opinion, the overall HDR efficiency is very high compared to previous studies (as also stated by reviewer 3). However, we will add additional experimental data on repeated editing of more targets to enrich for HDR editing events in a population of cells, as suggested by the reviewer. We will also add additional controls and corrections in the manuscript, as suggested by the reviewer.

Major comments:

- As a methods paper, the major shortcoming of this manuscript is that, while the authors show that

combined interference of DNA-PKcs and POLQ activity increases the purity of installing single-nucleotide variants into the genome, how well that finding will translate into practical utility remains unclear:

(1) The most useful version of their approach is what they call HDRobust (transient inhibition). How well cells will survive such treatment, however, is an open question. Indeed, loss of POLQ has been shown to be synthetic lethal with many DNA repair genes—including NHEJ factors (PMID: 31537809)—and the authors demonstrate that their mutant cells lines have strong cell growth phenotypes during editing (Supplementary Fig. 3). The same cell survival analysis should be performed and reported for HDRobust.

We will add experimental data on cell survival when using HDRobust.

(2) siRNAs are well known to have profound off-target effects (PMID: 20043028) and using five siRNAs to transiently inhibit POLQ is likely to be problematic for some applications (and is technically unnecessary). If RNAi-mediated depletion of POLQ is indeed the most robust strategy for this approach, a single siRNA that can perform well should be identified and described (and rescue of that effect by reintroduction of an siRNA-resistant POLQ should be demonstrated). Alternatively, small molecule inhibitors of POLQ's polymerase activity have been recently described (PMID: 34140467). Testing of these would be a valuable addition to this paper.

In our opinion, the genetic mutations in POLQ strongly supports that the effects seen are due to the inhibition of POLQ and not off-target effects.

In the HDRobust mix we opted to use a pool of siRNAs to achieve maximum HDR and outcome purity, but we already identified a single siRNA that performs well and achieves most of the effect (siRNA 765, Supp. Fig. 1a), if one prefers to use one siRNA only.

We thank the reviewer for the great suggestion to generate a siRNA resistant POLQ to exclude that the HDR-increasing effect is due to siRNA off-target binding. We will add experimental data for this. From this idea we are further inspired to generate POLQ variants with inactivated RAD51-binding sites (PMID 25642963) that should allow increased HDR-efficiencies if RAD51 sequestering by POLQ is indeed limiting for HDR.

Furthermore, we will test small molecule inhibitors of POLQ (ART558 and Novobiocin).

(3) One of the more exciting results in the paper is the finding that repeated editing with the authors' approach can be used to edit a bulk cell population to near uniformity “without the isolation of cellular clones” (Figure 2e). However, despite a broad claim that HDRobust is well suited for this purpose, this

particular feature is demonstrated at only one edit site (and without analysis of cell viability during editing). I'd recommend providing data to address (1) and (2), as well as expanding the examples of (3).

We will add additional experimental data for more targets employing repeated editing to enrich for HDR edited cells with high outcome purity in a population of cells.

- A second major concern is that the paper oversimplifies our current mechanistic understanding of DSB repair at nuclease induced double-strand breaks:

Example 1: The authors describe HDR as a single mechanistic process without discriminating between the forms of HDR they deploy in the paper (i.e., single-strand template repair, SSTR, and polymerase theta-mediated end-joining, TMEJ) and HDR mechanisms not evaluated (e.g., homologous recombination, HR). Ultimately, I'd argue that such oversimplifications do not necessarily diminish the actual results in the manuscript, but they do cause interpretation in the paper to be muddled and, at times, incorrect.

We will provide a more nuanced description of mechanistic processes involved in HDR that include single-strand template repair (SSTR) and homologous recombination (HR) in the introduction. We will also mention polymerase theta-mediated end-joining (TMEJ) as a specialized form of microhomology-mediated end-joining (MMEJ) and that MMEJ and HDR both require resected DSB ends with single stranded overhangs.

Most notably, it's important to point out that SSTR has been shown to be RAD51-independent in many studies, albeit not all (PMID: 16908537, 28067217, 33057349, 34672952). Given this observation, it is a glaring oversight for the authors to discuss RAD51 as definitive mediator of SSTR (aka "HDR") throughout. Of course, with this in mind, speculation "that the siRNA-induced cleavage of the mRNA" results "in a truncated proteins that may sequester RAD51 and thereby inhibit HDR" seems very unlikely in their experiments. I'd strongly recommend that the authors provide a more scholarly description of SSTR and either rescue the effect of their siRNA pool by reintroduction of an siRNA-resistant POLQ or provide a clear discussion outlining the caveats associated with inferring mechanism from the provided results.

We thank the reviewer for pointing this out. It is indeed interesting that the role for RAD51 in HDR using single-stranded donors is debated in the literature and we will add a discussion regarding inconsistencies in the literature with respect to dependence of RAD51 for HDR using single-stranded DNA donors.

In addition to the studies mentioned by the reviewer, we have previously observed (PMID 31392986) that a small molecule inhibitor of RAD51 (B02) reduces targeted nucleotide substitutions when using a single-stranded donor. Discrepancies in the literature might be due to that dependent on the cellular context, pathways that are distinct from homologous recombination and independent of RAD51 (e.g. fanconi anemia pathway) are also able to use a single-stranded donors for repair. Alternatively, if indeed all repair using single-stranded DNA donors would be independent of RAD51, a first DSB might be repaired using single-stranded DNA donors in a RAD51-independent fashion, while a second sequential DSB on the sister-chromatid might be repaired by classical RAD51-dependent HR using the edited chromatid as a template.

We will add additional experimental data employing the RAD51 inhibitor B02, and also generate a siRNA-resistant POLQ to investigate if the HDR-increasing effect of the siRNA is then lost. We thank the reviewer for this suggestion. This idea has also inspired us to generate POLQ variants with inactivated RAD51 binding sites (PMID 25642963). This should allow increased HDR-efficiencies if RAD51 sequestering by POLQ is indeed limiting for HDR.

Example 2: At least in yeast, SSTR is Rad52-dependent (PMID: 33057349). How the RAD52 mutations examined in the paper would be expected to impact SSTR is therefore not at all clear a priori. I'd strongly recommend that the authors provide a more nuanced description of published data regarding the potential role of Rad52 in SSTR and other DSB repair pathways.

We will provide a more nuanced description of published data regarding the role of RAD52 and DSB repair in the introduction.

- A third major concern centers on the authors' comparison of HDRobust to prime editing. The authors claim in their discussion that "editing using a DNA donor and HDRobust outperforms even optimized PE in terms of absolute precise editing efficiency (up to 91% HDR) as well as outcome purity (up to 96%)." This comparison is problematic for two reasons: First, an enormous amount of work has recently gone into optimizing prime editing systems (e.g., PMID: 34608327), but, rather than use the most optimized editors, the authors appear to have generated their own prime editing systems to facilitate comparisons.

We actually used chemically modified pegRNAs from PMID: 34608327 in our experiments and generated an iPrime cell line to be sure every cells expressed the prime editor so that we do not underestimate prime editing efficiencies if transfection efficiencies of plasmids would be suboptimal. Also we employed human codon optimization as done in PMID: 34653350. The most recent prime editor PE5 contains a dominant negative mismatch-mediated repair (MMR) protein (MLH1dn), which we did not include in our system as knockout of MLH1 did not increase prime editing efficiency for the tested *RNF2* target, which might be due to a different MMR response in embryonic stem cells (PMID: 25012654). We not not included these data as we thought they are out of the scope of this manuscript.

Moreover, some of the published prime editing rates provided in Figure 4b are clearly not from the most efficient nor the most precise method(s) of prime editing available (indeed, some even appear to have been selected from figures where better rates are included in the same figure, PMID: 34608327). In order to make a direct and fair comparison, authors must use state-of-the-art approaches/data.

As we did not use addition of MLH1dn (PE5) in our system and therefore used the published efficiencies of PE3 that are best comparable to our system. We will add efficiencies of improved epegRNAs using PE3 (PMID: 34608327) and for a full comparison we will now also state the efficiencies for PE5 from (PMID: 34653350). The additional efficiencies are in the range of the PE3 efficiencies in our first submitted manuscript (PMID 34608327: epegRNA *RNF2* target: PE3 ~55%; indels ~15%; epegRNA *FANCF* target: PE3 ~48%, indels 4%; PMID: 34653350: *RNF2* target: PE5 ~50%, indels ~ 10%; *FANCF* target: PE5 ~60%, indels 20%), except for the *CDKL5* target (*CDKL5* target: PE5: ~40%, indels ~10%) that has higher absolute efficiency than HDRobust, but lower outcome purity.

Thus, editing using a DNA donor and HDRobust outperforms even optimized PE (including PE5) in terms of outcome purity (up to 96%) and for two out of three targets in absolute precise editing efficiency (up to 91% HDR).

Minor comments:

- The abstract of this paper claims that “up to 91% of chromosomes in populations of cells will carry desired point mutations introduced by HDR” and the authors reiterate this “up to 91%” claim again in the discussion. In my view, such “up to” claims serve far less good than just reporting the actual range of editing observed.

In principle we agree that it is good practice to report the range. However, we do not want the reader to underestimate the power of the method if we would report 'from 14-91%'. The lower efficiencies like 14% are likely due to limitations of gRNA cleavage and suitability of DNA donor rather than HDR repair (e.g. Fig. 1C TTLL5, HDR is 14% with double inhibition, but total editing is also only 33% when no repair genes are inhibited). In addition, we will add data for more targets employing repeated editing to enrich for HDR edited cells with high outcome purity in a population of cells

We thus advocate to leave the sentence unchanged or write instead:

'We find that the combined inhibition of non-homologous end joining (NHEJ) and microhomology-mediated end joining (MMEJ) either by mutations or by inhibitory substances results in that HDR efficiencies are only limited by gRNA cleavage efficiency and suitability of DNA donor.'

- The diagrams indicating editing conditions (nucleases and guides) throughout the manuscript are confusing. Clarity would be improved by just using words to indicate conditions instead of cartoons.

We will add written descriptions in the figures throughout the manuscript to prevent confusion

- Recent work has shown that the presence of a single-strand oligo donor can impact TMEJ (PMID: 33160457, 34672952). Inclusion of no donor controls would therefore be a valuable addition to this paper.

We will add additional experimental data with no donor controls.

- For some reason, the authors relegate some key results to a Supplementary Discussion section rather than included them in the main text (e.g., effects on cell viability). I would recommend these data be included in the main text and figures.

We will move the cell survival discussion and figure to the main text as suggested by both reviewers 2 and 3.

Reviewer #3:

Remarks to the Author:

The submitted manuscript presents an impressive body of work aimed at a long-standing goal in the field of genome editing: maximizing a particular outcome from the DSB (or the nick, or a prime-edit) – the

introduction of a repair-template-specified mutation. A great wealth of studies have been devoted to this issue over the past 17 years, and it's safe to say that the kinds of numbers the authors are reporting in terms of HDR percentage are some of the best ever seen, and the work will be of interest to the large scientific community of genome editors.

We thank the reviewer for these encouraging words.

Comments / questions

The authors use a “nickase” form of Cas9 and two gRNAs to drive editing in “dual-nickase” mode. Do the authors have any data on single-nickase editing? As the authors are well-aware, there is a 10 year-long precedent (work from Sangamo and Keith Joung, among others; pubmed IDs 22434427 and 22373919) that a such a single-nickase approach markedly reduces non-HDR-driven outcomes.

While it is indeed true that single nicking can result in higher HDR purity, we did not attempt single nickase editing, since overall absolute editing efficiencies have been described to be much lower than for double nicking (PMID 28067217, 24529477, 23873081, 25398341, 27001513), suggesting the presence of a very efficient single-nick repair mechanism. For example, single nicking using Cas9 D10A resulted in 4% absolute overall editing with around 2% absolute HDR events (50% HDR outcome purity), while editing the same site with Cas9 resulted in 80% absolute overall editing, but only around 10% HDR outcome purity (Supp.Fig 7b in PMID 28067217).

It is unclear from the Materials and Methods whether any optimization of donor concentration was attempted. As the authors know, starting with the first report of using ssODNs for genome editing via HDR (2011 paper from Sigma-Aldrich, pubmed ID 21765410), it's been clear that titrating ODN quantity can improve the HDR/NHEJ ratio.

We use ssODN concentrations for both electroporation and lipofection such that doubling the concentration did not result in higher absolute HDR (PMID: 29867139) and refrained from using higher concentrations to limit toxicity induced by exogenous DNA. The optimized concentrations for our system are stated in the Methods section (200pmol for electroporation and 10nM for lipofection) and are comparable to those suggested from the ssODN vendor Integrated DNA Technologies.

It is unclear whether the authors introduced blocking mutations into the donor to prevent recleavage of the HDR-repaired template – another established way to limit NHEJ/MMEJ-based outcomes.

The DNA donor sequences can be found in the Supplementary Data and intended substitutions are bold. All donors were designed such that the mutation of interest also serves as a blocking mutation. To make this clear to the reader will add this information in the main text.

There is a striking disparity in Fig 1b between the first two genes and the second two. In the former, all the edited alleles convert to HDR ones (in other words, the absolute editing efficiency stays the same, and all the editing is resolved by HDR). For the second two, that is very clearly not the case. The absolute editing efficiency drops. Do the authors have any insight as to why?

We believe this could be due to I) perfect repair using the wild type sister chromatid by homologous recombination resulting in a high frequency of wild type sequences even though editing has occurred and/or II) cell death by pathway inhibition resulting in enrichment of non-transfected wild type cells.

Do the authors have any data on whether HDRobust affects hESC or iPSC stemness / totipotency? In other words, does the pharmacological inhibition of NHEJ and siRNA inhibition of SSA create an epigenetic memory that limits the ability to differentiate the stem cells into various progeny?

During preparation of the manuscript and while waiting for the reviewer's comments we have used HDRobust to introduce a substitution seen in Neandertals in the gene NOVA1 in human stem cells and successfully differentiated them into cerebral brain organoids – suggesting that differentiation capacity is not compromised after HDRobust treatment.

We would be happy to add these data to the manuscript if the editor and reviewer agree. This would add one figure and one supplementary figure to the paper. Below, is a summary of the results.

Summary of NOVA1 results:

Recently, Trujillo et al. introduced an ancestral mutation seen in Neandertals in the gene NOVA1 in stem cells. When they derived brain organoids from these cells they were of irregular shape and smaller than those derived from unedited cells (PMID: 33574182). We analyzed their sequencing data and found that the edited cells used by Trujillo et al. carried a deletion of the target site on one chromosome, suggesting

that this might be responsible for the phenotype (PMID: 34648345). This was disputed by a subset of authors of the original study (PMID: 34648331).

We have now investigated this by editing NOVA1 in 409B2 hiPSCs using HDRobust using the identical gRNA target and DNA donor as Trujillo et al. and either Cas9 RNP as used in the original study, or Cas9-HiFi RNP and HDRobust. Editing with Cas9-HiFi and HDRobust mix increased HDR-efficiency from 34% to 83% and reduced percentage of cellular clones with aberrant NOVA1 copy number from 69% to 3% compared to standard editing with Cas9. Subsequent organoid differentiation worked equally well in the wild-type and in the edited cell lines suggesting that the HDRobust does not affect the differential potential of the cells. Organoid shape and size are not affected by the ancestral NOVA1 mutation.

The cell survival discussion should be moved from supplementary to primary – it's an important issue. In particular the authors should draw the reader's attention to the fact that "dirty" guides will not work with HDRobust and instead the edited cells will just die.

We will move the cell survival discussion and figure to the main text as suggested by reviewers 2 and 3. In the discussion we plan to add a sentence to draw the reader's attention to the fact that "dirty" guides result in high cell death with HDRobust if high fidelity Cas9 or Cas9 double nicking are not used to prevent off-target cleavage.

Decision Letter, first revision:

Dear Stephan,

Thank you for your response to our queries on your manuscript "Predominant HDR-dependent genome editing and prevention of byproducts by DNA repair pathway choice" (NMEMETH-A49626B). I am pleased to report that we'll be happy in principle to publish it in Nature Methods, pending minor revisions to satisfy the referees' final requests and to comply with our editorial and formatting guidelines.

Please revise as per your point-by-point response to the reviewers. For the question about comparison to PE5, I suggest you keep the comparison with published efficiencies but state that a head-to-head comparison with PE5 was not performed. Please also adjust the claims as you have mentioned in your rebuttal.

TRANSPARENT PEER REVIEW

Nature Methods offers a transparent peer review option for new original research manuscripts submitted from 17th February 2021. We encourage increased transparency in peer review by publishing the reviewer comments, author rebuttal letters and editorial decision letters if the authors agree. Such peer review material is made available as a supplementary peer review file. Please state in the cover letter 'I wish to participate in transparent peer review' if you want to opt in, or 'I do not wish to participate in transparent peer review' if you don't. Failure to state your preference will result in delays in accepting your manuscript for publication.

ORCID

Sincerely,
Madhura

Madhura Mukhopadhyay, PhD
Senior Editor
Nature Methods

Reviewer #1 (Remarks to the Author):

Riesenberg et al in their revised version of the manuscript entitled "Predominant HDR-dependent genome editing and prevention of byproducts by DNA repair pathway choice" describe their work in altering genome editing outcomes by perturbing DNA-PKcs and Pol-theta based repair. They use both genetic perturbations, siRNA perturbations and small molecule perturbations. Their results demonstrate that across a variety of loci and variety of cell types that dual inhibition biases repair of a DSB to an HDR outcome. They use a term called "genome editing purity" to quantify their results. The major concern, that the authors did not fully address, is what happens to the breaks (both on and off-target) that are unrepaired. The perfect repair breaks seem to disappear even though breaks must still be occurring. In conjunction with the loss of viability and cell number with double inhibition, this raises concerns that there is significant cell drop out. As a research method to more efficiently generate cell lines, this might be useful but as a method for gene therapy, even in cell manipulation, there remain important genotoxic concerns. Thus, my strong recommendation is that the authors in the discussion emphasize that given the current results, it could be a useful approach for disease modeling and research but that the approach is not validated yet for clinical development. The work does add a new tool to the genome editing toolbox and gives scientists an opportunity to choose to use it or not. The direct comparison to Prime editing is of value to the community.

Remaining concerns and comments are as follows:

1. I disagree with the authors rebuttal that it is rigorous to say "up to..." in the abstract. They should give a median with SEM or range.
2. As noted above, the authors have not addressed the fate of cells with unrepaired breaks. They certainly could be more quantitative about the toxicity.
3. the toxicity without a template is important and needs to be identified in the abstract. This approach should be used only for template based HDR.
4. While the authors in the rebuttal discuss including references to AAV templated HDR, they did not seem to include those references and the results of HDR using AAV in the introduction or the discussion. The manuscript has to be corrected in this regards. Very high frequencies of HDR have been achieved in a wide variety of cell types, including IPS cells using the AAV system. The authors are misleading the readers by not including this literature.
5. It is increasingly recognized that guides have more "NHEJ like INDELS" or "MMEJ like INDELS." The authors should describe the baseline spectrum for each gRNA before pathway inhibition. They also do not display INDEL spectrums in a standard way and the histogram plots are not as informative as simply listing the INDELS found and their frequency. ICE and TIDE give that output.

6. What was the frequency of large on-target deletions using the method described by Park et al in Science Advances? Did it change with using the dual pathway inhibition?
7. The authors have inadvertently developed a screen for gRNAs with low specificity as it seems those guides have high toxicity when combined with dual inhibition. This could easily be turned into a functional screen for low-specificity guides without actually having to identify the potential off-target sites.
8. I don't believe the organoid work adds value without quantifying number of organoids without and with dual inhibition.

Reviewer #2 (Remarks to the Author):

Reviewer #2:

Overall, the manuscript by Riesenberg and colleagues is much improved.

The authors have done a good job addressing concerns about the practical utility of the approach, especially with regard to cell viability. That said, it would be useful to know how long after editing the viability measurements were taken, and I would encourage them to include the cell viability data from S7b in the main text. While ~60% viability is more than I would have expected, it's still likely to be a deal breaker for some applications.

I appreciate that the authors took the time to provide a more accurate description and interpretation of repair mechanisms. Thank you.

The only outstanding concerns I have are:

(1) The comparison to prime editing. The authors maintain their claim that "HDRobust performs better than prime editing in terms of absolute precise editing efficiency" but still do not compare the method to the most optimized version of prime editing. In their response, the authors state that they did not perform this comparison because "knockout of MLH1 did not increase prime editing efficiency for the tested RNF2 target". As far as I can tell, lack of improvement at the RNF2 site in the cited work applies to a G-to-C substitution. An important consideration here is that G-to-C edits are less sensitive to mismatch repair inhibition () and so knockout of MLH1 wouldn't be expected to have a large effect on that

particular edit (PMID: 36646933). By contrast, the site tested in by Riesenber et al is a C-to-A mutation, and thus is likely to be responsive, especially in MMR-proficient cells (i.e., not HEK329Ts). Consistent with this, other RNF2 edits appear to be improved by PE5 within the cited work. Regardless of this technicality, the larger point here is that while head-to-head comparisons of state-of-the-art methods can be useful, broad claims of method superiority from a small number of tests are typically counterproductive to the field as a whole. Given that, for this paper, I would recommend just adjusting the claims within the paper to match the presented data; e.g., “HDRobust outperforms PE3 by metrics of purity and efficiency, although comparisons to enhanced prime editing methods remain to be performed, and when selecting an editing method, other metrics (e.g., viability of a given cell model) may be important to consider as well.”

(2) Inclusion of an “up to 91%” claim in the abstract and discussion. Independent of the particular variables that limit the lower end of editing, the range observed is the range observed. The high end of this range is impressive all on its own and I think that reporting a full range may be really helpful to those who will be looking to apply the method. But, this is ultimately a style issue, and I completely understand why authors chose such wording.

Reviewer #3 (Remarks to the Author):

The revised manuscript comprehensively addresses comments raised during initial review.

Author Rebuttal, first revision:

Editors Comments:

In particular, in line with Ref2's concern, we agree that benchmarking against PE5 must be added.

We already compared our efficiencies against published PE5 efficiencies from the world-leading prime editing labs using the same genomic targets. See the comments to reviewer 2.

We are happy to follow the suggestion of reviewer 2 to adjust the claims in the Discussion exactly as s/he suggests. This would allow swift revision of the manuscript.

As we point out in our answer to Reviewer 2, a direct experimental comparison to PE5 in our system would take two or more additional months in order to optimize PE5 in stem cells. In the end, it would still not be clear if PE5 would be as efficient in our hands as in the hands of the inventors of the method. We therefore believe that a comparison of the published efficiencies of the same targets adequately serves the purpose of comparing the two methods.

If there are requests that you feel are inappropriate or would require unreasonable experimental effort to

address, please fully explain your arguments. This will help us as editors to make a more informed decision on your manuscript.

We are thankful for the reviewers' suggestions and have incorporated most of them. Below, we explain, why a few suggestions are not feasible or would require time and resources without adding much more value to the manuscript. See especially the discussion of comparison to PE5 above and in the response to Reviewer 2.

Reviewers' Comments:

Reviewer #1:

Remarks to the Author:

Riesenberg et al in their revised version of the manuscript entitled "Predominant HDR-dependent genome editing and prevention of byproducts by DNA repair pathway choice" describe their work in altering genome editing outcomes by perturbing DNA-PKcs and Pol-theta based repair. They use both genetic perturbations, siRNA perturbations and small molecule perturbations. Their results demonstrate that across a variety of loci and variety of cell types that dual inhibition biases repair of a DSB to an HDR outcome. They use a term called "genome editing purity" to quantify their results.

The major concern, that the authors did not fully address, is what happens to the breaks (both on and off-target) that are unrepaired. The perfect repair breaks seem to disappear even though breaks must still be occurring. In conjunction with the loss of viability and cell number with double inhibition, this raises concerns that there is significant cell drop out.

The data strongly suggests that cells with breaks (both on and off-target) that are unrepaired simply die, likely due to DNA-damage induced apoptosis. The only way to repair a break under dual inhibition is to use the exogenous DNA donor or the homologous chromosome. We summarize this in the Discussion: "Because dual inhibition of NHEJ and MMEJ prevents copy number loss and off-target editing (Fig. 3, 4), we speculate that cells lacking these end-joining pathways cannot repair DSBs in the absence of a suitable DNA donor, resulting in that they can only repair breaks using the exogenous DNA donor or sister chromatids at the on-target site, and solely sister chromatids at off-target sites. When DSBs cannot be repaired in one of these ways, especially when excessive off-target cleavage occurs, cells will die resulting in a population precisely edited and wild-type cells. Consistent with this, cells will die or remain wild-type when no DNA donor is provided (Extended Data Fig. 3)."

As a research method to more efficiently generate cell lines, this might be useful but as a method for gene therapy, even in cell manipulation, there remain important genotoxic concerns. Thus, my strong recommendation is that the authors in the discussion emphasize that given the current results, it could be a useful approach for disease modeling and research but that the approach is not validated yet for clinical development.

We added a sentence in the discussion emphasizing this.

The work does add a new tool to the genome editing toolbox and gives scientists an opportunity to choose to use it or not. The direct comparison to Prime editing is of value to the community.

We are thankful for the appreciation of the comparison to prime editing.

Remaining concerns and comments are as follows:

1. I disagree with the authors rebuttal that it is rigorous to say "up to..." in the abstract. They should give a median with SEM or range.

As also suggested by reviewer 2, we now not only state the maximum efficiency but also added the median of all targets with SEM.

2. As noted above, the authors have not addressed the fate of cells with unrepaired breaks. They certainly could be more quantitative about the toxicity.

See first comment to reviewer 1. We added substantial data regarding cell survival for the revision. Reviewer 2 explicitly acknowledged the improvement of the manuscript with regard to concerns on cell viability.

3. the toxicity without a template is important and needs to be identified in the abstract. This approach should be used only for template based HDR.

We agree that the approach of NHEJ+MMEJ double inhibition should and can only be used for template based HDR. The title and abstract clearly state the intent to apply HDR editing, but we now also added the wording 'template based HDR' to the abstract.

4. While the authors in the rebuttal discuss including references to AAV templated HDR, they did not seem to include those references and the results of HDR using AAV in the introduction or the discussion. The manuscript has to be corrected in this regards. Very high frequencies of HDR have been achieved in a wide variety of cell types, including IPS cells using the AAV system. The authors are misleading the readers by not including this literature.

As pointed out in the response letter, we had incorporated the following sentence in the Discussion: "Although not tested here, HDRobust is likely to also increase outcome purity and further increase already high efficiencies achieved when using AAV6 donors that hold great promise for therapeutic gene editing (PMID: 30082871, 34703842, 35617958, 34086870)."

5. It is increasingly recognized that guides have more "NHEJ like INDELS" or "MMEJ like INDELS." The authors should describe the baseline spectrum for each gRNA before pathway inhibition. They also do not display INDEL spectrums in a standard way and the histogram plots are not as informative as simply listing the INDELS found and their frequency. ICE and TIDE give that output.

Most studies only plot the sum of indels. ICE and TIDE utilize Sanger sequencing data while we use Illumina sequencing. We believe our re-categorization of indels based on insertions and deletion with various lengths of microhomology is sufficiently descriptive as the main focus is on improving HDR. Nevertheless, we now added an additional Supplementary file with CRISPResso cut site plots (see below) for the Cas9 gRNAs for which we have compared two different methods to distinguish NHEJ and MMEJ like indels (Extended Data Fig. 6) to further illustrate the differences of indel patterns. For researchers interested in even deeper analysis of indels, all sequencing data is available in a public repository.

CRISPResso cut site plot

6. What was the frequency of large on-target deletions using the method described by Park et al in Science Advances? Did it change with using the dual pathway inhibition?

We did not cite this paper and believe the reviewer refers to PMID: 36269834. They find large deletions at Cas9 on-target cut sites on the HBB (11.7 to 35.4%), HBG (14.3%), and BCL11A (13.2%) genes in HSPCs and the PD-1 (15.2%) gene in T cells. Park et al. did not employ any pathway inhibition so we are unsure what the reviewer intended with the question.

In our study we reduce hESC clones with on-target copy number changes for SCAP (8.3%) and TEX2 (13%) to 0% when combined with dual pathway inhibition by HDRobust. HDRobust also reduced the number of hiPSC clones with on-target copy number changes for NOVA1 from 69% to 3%.

7. The authors have inadvertently developed a screen for gRNAs with low specificity as it seems those guides have high toxicity when combined with dual inhibition. This could easily be turned into a functional screen for low-specificity guides without actually having to identify the potential off-target sites.

We have already started a related project and are encouraged that the reviewer acknowledges the potential use of such a tool. We added a sentence in the discussion that states the possibility to use double inhibition for a functional screen for low-specificity guides without actually having to identify the potential off-target.

8. I don't believe the organoid work adds value without quantifying number of organoids without and with dual inhibition.

The organoid work shows that HDRobust does not compromise the ability of stem cells to differentiate into brain organoids. We believe a number of readers will share reviewer 3' interest in differentiation ability after HDRobust treatment.

Reviewer #2:

Overall, the manuscript by Riesenbergs and colleagues is much improved.

The authors have done a good job addressing concerns about the practical utility of the approach, especially with regard to cell viability. That said, it would be useful to know how long after editing the viability measurements were taken, and I would encourage them to include the cell viability data from S7b in the main text. While ~60% viability is more than I would have expected, it's still likely to be a deal breaker for some applications.

The cell viability measurements were taken three days after editing. We write this in the methods in the 'resazurin assay' section: "Subsequent to editing, cells were grown in media containing ROCK inhibitor Y-27632 for one day, followed by normal media for two days before being supplied with fresh media containing 10% resazurin solution"

We write in the Results in the 'Transient repair pathway inhibition' section: "Editing without DNA donors resulted in 75% to 94% cell survival (mean 87%), while editing using DNA donors and HDRobust resulted in a wider range of 35% to 96% (mean 59%) (Extended Data Fig. 7)."

I appreciate that the authors took the time to provide a more accurate description and interpretation of repair mechanisms. Thank you.

We also want to thank the reviewer for the suggestions that clearly improved the paper.

The only outstanding concerns I have are:

(1) The comparison to prime editing. The authors maintain their claim that "HDRobust performs better than prime editing in terms of absolute precise editing efficiency" but still do not compare the method to the most optimized version of prime editing. In their response, the authors state that they did not perform this comparison because "knockout of MLH1 did not increase prime editing efficiency for the tested RNF2 target". As far as I can tell, lack of improvement at the RNF2 site in the cited work applies to a G-to-C substitution. An important consideration here is that G-to-C edits are less sensitive to mismatch repair inhibition () and so knockout of MLH1 wouldn't be expected to have a large effect on that particular edit (PMID: 36646933). By contrast, the site tested in by Riesenbergs et al is a C-to-A mutation, and thus is likely to be responsive, especially in MMR-proficient cells (i.e., not HEK329Ts). Consistent with this, other RNF2 edits appear to be improved by PE5 within the cited work. Regardless of this technicality, the larger point here is that while head-to-head comparisons of state-of-the-art methods can be useful, broad claims of method superiority from a small number of tests are typically counterproductive to the field as a whole. Given that, for this paper, I would recommend just adjusting the claims within the paper to match the presented data; e.g., "HDRobust outperforms PE3 by metrics of purity and efficiency, although comparisons to enhanced prime editing methods remain to be performed, and when selecting an editing method, other metrics (e.g., viability of a given cell model) may be important to consider as well."

We actually did apply the C-to-A mutation for RNF2 in hESCs that contained a stop codon in MLH1 and this did not increase prime editing efficiency, which is why we did not test other targets and refrained from including it in the manuscript. In the response letter we then referenced a paper which did not apply prime editing, but investigated different MMR responses in embryonic stem cells (PMID: 25012654). This was to give a potential explanation of the lack of improvement by the MLH1 knockout in our hands in hESCs.

In the Discussion, we thus benchmark our efficiencies to published efficiencies of using enhanced prime editing methods (epegRNAs and PE5) and wrote that: “editing using a DNA donor and HDRobust outperforms even further optimized PE (PMID: 34608327 epegRNAs, PMID: 34653350 PE5 = PE3 + MLH1dn) in terms of outcome purity (up to 96%) and for two out of three targets in absolute precise editing efficiency (up to 91% HDR) (Fig. 5a, b, c).” We think this is a fair comparison since the cited labs are at the forefront of prime editing and they are likely to arguably achieve the highest prime editing efficiencies in the community.

Direct comparison using the published PE5 plasmids would likely make prime editing appear very inefficient, because using PE3 plasmids we (similar to many other labs, personal communication) could at best achieve low single digit prime efficiencies. This would likely translate to PE5 plasmids in our hands/cell type. A good experimental comparison could be to use the same gRNA as used in PMID: 34653350 (PE5 paper) to make a MLH1 mutant with the identical published frameshift in hESCs and then do similar edits as done for iPrime PE3 in Fig. 5a. However, this would likely take several months to generate the cell line and do the edits.

To avoid additional experiments that on the one hand would likely only recapitulate published PE5 efficiencies or result in lower efficiencies and on the other hand cost valuable additional time and resources, we adjusted the claims by inserting the sentence suggested by Reviewer 2: “HDRobust outperforms PE3 by metrics of purity and efficiency, although comparisons to enhanced prime editing methods remain to be performed, and when selecting an editing method, other metrics (e.g., viability of a given cell model) may be important to consider as well.”

(2) Inclusion of an “up to 91%” claim in the abstract and discussion. Independent of the particular variables that limit the lower end of editing, the range observed is the range observed. The high end of this range is impressive all on its own and I think that reporting a full range may be really helpful to those who will be looking to apply the method. But, this is ultimately a style issue, and I completely understand why authors chose such wording.

As also suggested by reviewer 1, we added the median and SEM for all targets.

Reviewer #3:

The revised manuscript comprehensively addresses comments raised during initial review.

We are thankful for the appreciation of our work.

Final Decision Letter:

Dear Stephan,

I am pleased to inform you that your Article, "Predominant HDR-dependent genome editing and prevention of byproducts by DNA repair pathway choice", has now been accepted for publication in Nature Methods. Your paper is tentatively scheduled for publication in our August print issue, and will be published online prior to that. The received and accepted dates will be Jun 21, 2022 and Jun 12,

2023. This note is intended to let you know what to expect from us over the next month or so, and to let you know where to address any further questions.

Once your paper is typeset, you will receive an email with a link to choose the appropriate publishing options for your paper and our Author Services team will be in touch regarding any additional information that may be required.

Please note that *Nature Methods* is a Transformative Journal (TJ). Authors may publish their research with us through the traditional subscription access route or make their paper immediately open access through payment of an article-processing charge (APC). Authors will not be required to make a final decision about access to their article until it has been accepted. [Find out more about Transformative Journals](https://www.springernature.com/gp/open-research/transformative-journals)

Your paper will now be copyedited to ensure that it conforms to Nature Methods style. Once proofs are generated, they will be sent to you electronically and you will be asked to send a corrected version within 24 hours. It is extremely important that you let us know now whether you will be difficult to contact over the next month. If this is the case, we ask that you send us the contact information (email,

phone and fax) of someone who will be able to check the proofs and deal with any last-minute problems.

If, when you receive your proof, you cannot meet the deadline, please inform us at rjsproduction@springernature.com immediately.

Once your manuscript is typeset and you have completed the appropriate grant of rights, you will receive a link to your electronic proof via email with a request to make any corrections within 48 hours. If, when you receive your proof, you cannot meet this deadline, please inform us at rjsproduction@springernature.com immediately.

Once your paper has been scheduled for online publication, the Nature press office will be in touch to confirm the details.

Once your paper has been scheduled for online publication, the Nature press office will be in touch to confirm the details.

Content is published online weekly on Mondays and Thursdays, and the embargo is set at 16:00 London time (GMT)/11:00 am US Eastern time (EST) on the day of publication. If you need to know the exact publication date or when the news embargo will be lifted, please contact our press office after you have submitted your proof corrections. Now is the time to inform your Public Relations or Press Office about your paper, as they might be interested in promoting its publication. This will allow them time to prepare an accurate and satisfactory press release. Include your manuscript tracking number NMETH-A49626C and the name of the journal, which they will need when they contact our office.

About one week before your paper is published online, we shall be distributing a press release to news organizations worldwide, which may include details of your work. We are happy for your institution or funding agency to prepare its own press release, but it must mention the embargo date and Nature Methods. Our Press Office will contact you closer to the time of publication, but if you or your Press Office have any inquiries in the meantime, please contact press@nature.com.

Nature Portfolio journals [encourage authors to share their step-by-step experimental protocols](https://www.nature.com/nature-research/editorial-policies/reporting-standards#protocols) on a protocol sharing platform of their choice. Nature Portfolio 's Protocol Exchange is a free-to-use and open resource for protocols; protocols deposited in Protocol Exchange are citable and can be linked from the published article. More details can found at www.nature.com/protocolexchange/about.

Please note that you and any of your coauthors will be able to order reprints and single copies of the issue containing your article through Nature Portfolio 's reprint website, which is located at <http://www.nature.com/reprints/author-reprints.html>. If there are any questions about reprints please send an email to author-reprints@nature.com and someone will assist you.

Best regards,
Madhura

Madhura Mukhopadhyay, PhD
Senior Editor
Nature Methods